# When, Where and Why to Average Weights?

**Niccolò Ajroldi** [1 2]   **Antonio Orvieto** [1 2 3]   **Jonas Geiping** [1 2 3]

## Abstract

Averaging checkpoints along the training trajectory is a simple yet powerful approach to improve the generalization performance of Machine Learning models and reduce training time. Motivated by these potential gains, and in an effort to fairly and thoroughly benchmark this technique, we present an extensive evaluation of averaging techniques in modern Deep Learning, which we perform using AlgoPerf (Dahl et al., 2023), a large-scale benchmark for optimization algorithms. We investigate whether weight averaging can reduce training time, improve generalization, and replace learning rate decay, as suggested by recent literature. Our evaluation across seven architectures and datasets reveals that averaging significantly accelerates training and yields considerable efficiency gains across all considered workloads, at the price of a minimal implementation and memory cost, while mildly improving generalization. Finally, we explore the relationship between averaging and learning rate annealing and show that combining the two achieves optimal performance.

## 1. Introduction

Training Deep Learning models is both resource-expensive and time-consuming. A principled and simple approach to speed up training, dating back to Polyak (1990) and Ruppert (1988), involves averaging the model's weights across training iterations. This can either be performed online during training or post hoc by averaging checkpoints, effectively creating an ensemble of models at minimal additional cost. Previous studies have shown that weight averaging (WA) techniques can improve generalization (Merity et al., 2017; Gupta et al., 2020; Kaddour, 2022; Melis, 2023), increase robustness (Morales-Brotons et al., 2024), smooth loss landscapes (Izmailov et al., 2019), and accelerate convergence

(Athiwaratkun et al., 2018; Li et al., 2023; Sanyal et al., 2023). Recent studies have also explored the connection between learning rate decaying and weight averaging (Sandler et al., 2023; Hägele et al., 2024), and used the latter to develop schedule-free algorithms (Defazio et al., 2024b).

Table 1: Estimated training cost for one run on the AlgoPerf collection of models and tasks. Even for industrial-scale tasks like benchmark workloads, weight averaging reliably reduces compute costs.

|  | NadamW | +LAWA | +EMA |
| --- | --- | --- | --- |
| GPU-Hours | 612 | 550 | 541 |

In this work, we present the largest evaluation of averaging techniques in modern Deep Learning, which we perform using AlgoPerf (Dahl et al., 2023), a collection of large-scale workloads and architectures developed to provide a unified benchmark for optimization algorithms. Framing our analysis in this setting provides a carefully designed evaluation framework, enables comparisons against strong, heavily tuned baselines, and allows drawing broad and robust conclusions.

Building on previous work, we investigate the following questions. (i) Can weight averaging *reduce training time* across multiple models and tasks? (ii) Does averaging checkpoints *improve the generalization* performance of existing optimization algorithms? (iii) Is weight averaging merely a proxy for a shorter learning rate decay schedule, and can it fully replace learning rate decay?

Our contributions are as follows.

1. We show that averaging can significantly accelerate training dynamics across seven different models and architectures and estimate a 12% reduction in GPU-hours to train the entire AlgoPerf suite up to the validation target compared to our baseline. We find this effect to be consistent across hyperparameters and observe encouraging speed-ups even on a more sophisticated optimizer like Distributed Shampoo (Shi et al., 2023).

---

[1]ELLIS Institute Tübingen [2]Max Planck Institute for Intelligent Systems, Tübingen, Germany [3]Tübingen AI Center. Correspondence to: Niccolò Ajroldi <niccolo@tue.ellis.eu>.

*Proceedings of the 42nd International Conference on Machine Learning*, Vancouver, Canada. PMLR 267, 2025. Copyright 2025 by the author(s).

2. Beyond efficiency gains, we demonstrate how averaging achieves improved generalization across all the considered workloads, and show that combining WA with learning rate annealing yields optimal results.

3. Finally, we demonstrate how averaging checkpoints can act as a proxy for a shorter learning rate decay, but that it *cannot* fully replace learning rate schedules, at least within the explored variants, addressing an important question about the role of these techniques (Hägele et al., 2024; Defazio et al., 2024b).

The paper is organized as follows: Section 2 reviews related work on weight averaging; Section 3 discuss the methodology and experimental setup; Section 4, 5, 6 and 7 present our findings; Section 8 addresses the limitation of our analysis and Section 9 concludes.

## 2. Related Work

The idea of averaging iterates along a stochastic optimization trajectory dates back to Polyak (1990) and Ruppert (1988), and is often referred to as Polyak–Ruppert averaging. It has been extensively studied in the stochastic approximation framework (Polyak & Juditsky, 1992; Bach & Moulines, 2013; Neu & Rosasco, 2018; Lakshminarayanan & Szepesvari, 2018), and is a common technique to derive convergence guarantees in both convex and nonconvex optimization (Garrigos & Gower, 2023).

**Deep Learning applications.** Weight averaging techniques have seen extensive use in the Deep Learning community, often without explicit recognition or emphasis, despite their effectiveness. Influential works such as Szegedy et al. (2016), Vaswani et al. (2017), and Merity et al. (2017) have demonstrated their ability to enhance model performance and mitigate overfitting.

**SWA.** The work of Izmailov et al. (2019) sparked renewed interest in weight averaging by demonstrating how averaging points along the SGD trajectory leads to wider minima and improves generalization performance. Their approach, Stochastic Weight Averaging (SWA), has since been applied to semi-supervised learning (Tarvainen & Valpola, 2018; Athiwaratkun et al., 2018), low-precision training (Yang et al., 2019), domain generalization tasks (Cha et al., 2021; Arpit et al., 2022), and meta-optimization (Li et al., 2023).

**LAWA and EMA.** In the original formulation of SWA, a pretrained model is trained with a cyclical or constant learning rate, and multiple checkpoints are collected and later averaged. Kaddour (2022) proposed Latest Weight Averaging (LAWA), an online algorithm that averages the latest checkpoints in a rolling window, showing significant speedups on vision and language tasks. Further modifications of LAWA demonstrated notable boosts in pretraining modern

decoder-only language models (Sanyal et al., 2023). A valuable alternative to moving window averaging techniques like LAWA is Exponential Moving Averaging (EMA) (Li et al., 2024; Morales-Brotons et al., 2024; Arpit et al., 2022). It retains similar advantages of rolling window averaging, and constitutes an indispensable technique for high-quality image synthesis models such as GANs and diffusion models (Yazıcı et al., 2019; Song et al., 2021; Karras et al., 2024).

**Connection to learning rate annealing.** Classical stochastic smooth convex optimization rates showcase a tight link between WA and learning rate annealing, suggesting a practical interplay between these techniques (see e.g. Theorem 5.3. in Garrigos & Gower (2023)). Intuitively, averaging weights along the training trajectory reduces noise and might act as a proxy for learning rate decay. In fact, Sandler et al. (2023) proved the theoretical and empirical equivalence between WA and decaying learning rate for SGD. However, despite this appealing result, modern Deep Learning models are still predominantly trained with learning rate annealing (Hägele et al., 2024), even when maintaining an EMA of model weights (DeepSeekAI, 2025). A recent study by Defazio et al. (2024b) specifically investigates this connection, proposing an approach that fully replaces learning rate schedules with iterate averaging and demonstrating strong performance on the same benchmark used in this analysis. Whereas Defazio et al. (2024b) incorporates averaging directly into the optimization procedure, we explore a different flavor of averaging, where the averaged weights do not influence the updates—akin to Polyak averaging, SWA, LAWA, and EMA.

**Model soups.** Finally, a different but notable approach that leverages the benefits of averaging is model soups (Wortsman et al., 2022). In this case, multiple models are trained with different hyperparameter configurations and later aggregated, resulting in an ensemble with improved accuracy and robustness.

In this work, we demonstrate the benefits of weight averaging techniques on a challenging optimization benchmark (Dahl et al., 2023), hoping to encourage broader adoption of these methods in training large-scale Machine Learning models.

## 3. Experimental Setup

**Comparing optimization algorithms.** When evaluating optimization algorithms, two strategies are possible:

(A) Fixing a challenging target loss value and comparing the runtime needed to reach it.

(B) Comparing generalization performance within a fixed budget, either specified in number of steps or wall-clock time.

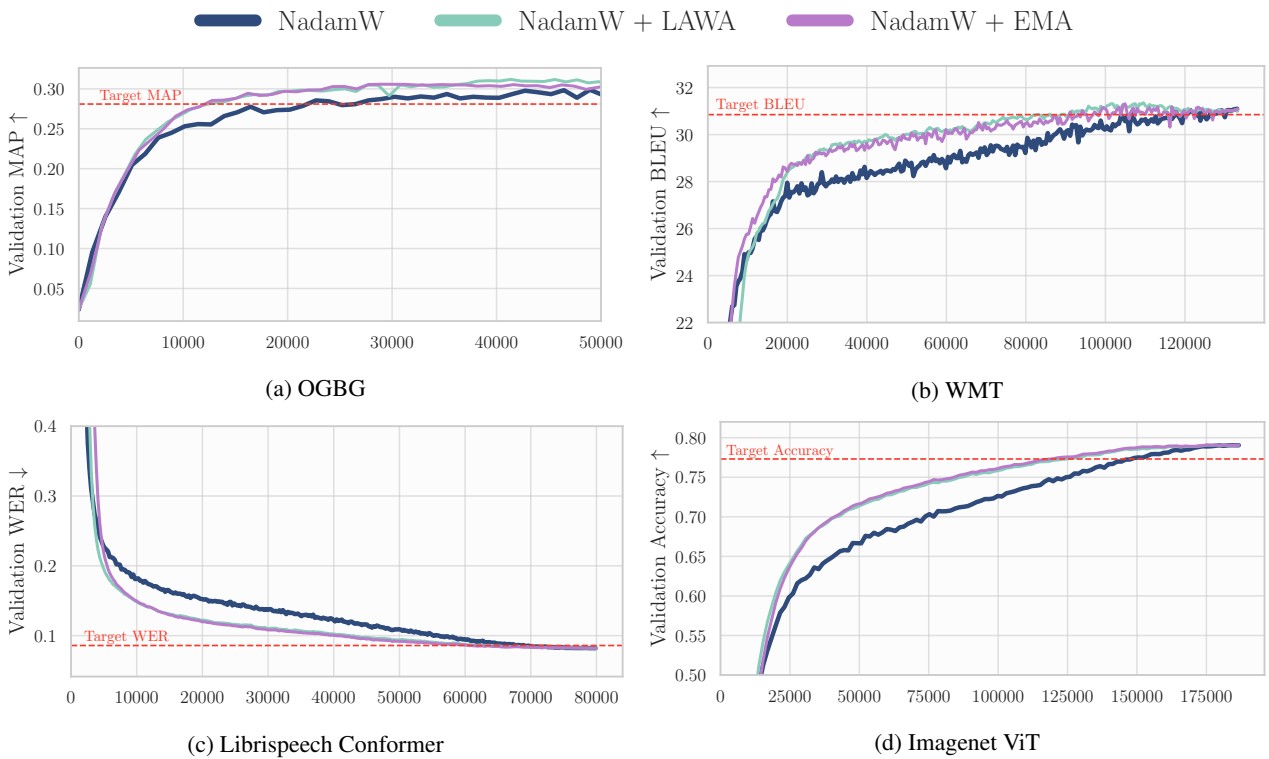

Figure 1: Weight averaging speeds up training across all considered workloads. The averaged schemes consistently achieve better performance during training, reaching the validation score target faster than the baseline algorithm. We display the validation score against the number of iterations (optimization steps) across different workloads; the dotted line represents the target score on each workload.

In this work, we investigate whether weight averaging improves existing optimization algorithms in *either* of the two described frameworks.

**Weight averaging.** We explore two flavors of weight averaging discussed in Section 2: LAWA (Kaddour, 2022) and EMA. For both approaches, we update a buffer that stores previous information about the model history. In the first case, we update a circular queue of length $L$, which stores recent checkpoints, while in the latter case, we maintain an exponential moving average of the model parameters, with coefficient $\gamma$. We update the averaging buffer every $\nu$ iterations, and explore the effect of such hyperparameter together with the value of $L$ and $\gamma$. In most scenarios, we save checkpoints of the baseline algorithm and run averaging schemes offline, but when testing collecting consecutive checkpoints, we use online versions of LAWA and EMA to avoid excessive disk storage. We discuss implementation details of the algorithms, along with their practical implications and considerations, in Appendix E.

**AlgoPerf.** We conduct our analysis on AlgoPerf (Dahl et al., 2023), a large suite of Deep Learning workloads, which provides a comprehensive benchmark for testing op-

timization algorithms. The benchmark is composed of eight *workloads*, each defined by a *dataset*, a *model* architecture, and a predefined *target metric* on a held-out set, designed to represent optimal performance on such a workload. We note that AlgoPerf has been developed following option (A) and that significant effort has been dedicated to deriving challenging target scores. Thus, it provides an ideal setting to evaluate the impact of weight averaging techniques applied to strong optimization algorithms and compare their performance against heavily tuned baselines. We also make use of the same datasets and architectures to score algorithms by means of their generalization capabilities (when following option (B)). We consider the following workloads from the AlgoPerf suite: (i) a DLRMsmall model on Criteo 1TB dataset for click-through rate prediction; (ii) U-Net on FastMRI for medical image reconstruction; (iii) ViT on ImageNet-1k for image classification; (iv) a GNN model on OGBG for graph-based prediction; (v) a Transformer-Big on WMT for machine translation; (vi) a Conformer for speech recognition; (vii) a DeepSpeech model on LibriSpeech for speech-to-text. We exclude ImageNet-ResNet from this analysis because, as shown in Dahl et al. (2023), no optimization algorithm successfully solves this task alongside

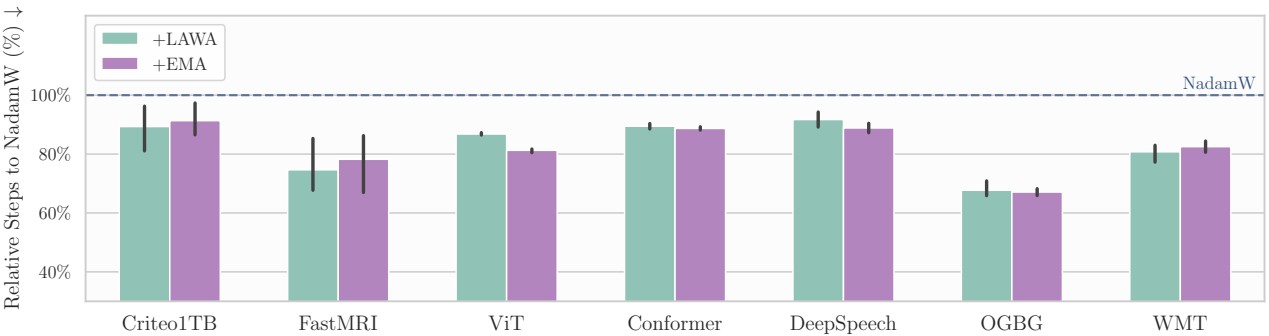

Figure 2: LAWA and EMA speed up training across several architectures and datasets. Both averaging schemes consistently outperform the baseline, achieving on average the benchmark target score using 82% of the steps required by NadamW. We estimate a 12% reduction in GPU-hours to train the entire AlgoPerf suite of workloads with respect to NadamW.

the other workloads, leaving no baseline on which to build. We refer to Appendix A for more details on the workloads.

**Baseline optimizers.** We build on top of existing optimization algorithms that perform well on AlgoPerf, enhancing them with LAWA or EMA. Unless otherwise specified, we use NadamW (Timothy, 2016; Loshchilov & Hutter, 2017a) as our baseline and name the resulting algorithms NadamW + LAWA and NadamW + EMA, respectively. Additionally, we investigate the potential benefits of combining a higher-order optimizer, such as Distributed Shampoo (Shi et al., 2023), with these averaging methods, exploring how this approach might further improve training efficiency. Given the high computational cost of the benchmark, we conduct most of the analysis using the best performing hyperparameters in Dahl et al. (2023) and Kasimbeg et al. (2025): this gives us strong baselines, avoiding the burden of expensive hyperparameter tuning. Additionally, we ablate on the role of the learning rate in each analysis and study in more detail the consequences of changing the learning rate schedule. Given this reference algorithm, we add LAWA or EMA on top of it, tuning only the hyperparameters of the averaging scheme and leaving the other baseline hyperparameters fixed, including learning rate schedule and weight decay. Unless otherwise specified, or when explicitly ablating on it, we train using a cosine learning rate schedule (Loshchilov & Hutter, 2017b).

**Optimal averaging horizon.** The impact of an averaging scheme largely depends on the *horizon* over which it is applied and on the *frequency* at which checkpoints are collected. A long horizon may prioritize outdated model checkpoints and generalize worse, whereas a short one may be suboptimal or result in a serious computational overhead. At the same time, updating the averaging buffers every step is only possible in the online version of LAWA and EMA, and collecting checkpoints too rarely might reduce the benefits of averaging. Previous works have collected checkpoints at each training step (Morales-Brotons et al., 2024; Hägele

et al., 2024), or the end of each epoch (Kaddour, 2022), and investigated the optimal horizon and update frequency on a single task (Sanyal et al., 2023). We explore combinations and interactions of these variables across architectures and objectives, and compare LAWA with EMA.

To control for variability in model initialization and data shuffling, we repeat experiments for three different seeds and report the mean and standard deviation.

## 4. Speeding up Training

We ask whether equipping a strong baseline with LAWA or EMA can reduce the number of steps required to reach a predefined validation target (option A in Section 3). To evaluate this, we track the *number of steps* required to reach the benchmark score on the held-out set. The target scores, derived in Dahl et al. (2023) by heavily tuning and comparing 4 different optimization algorithms, represent desirable performance on each workload.

**Efficiency gains of averaging.** When looking for a scheme that reduces training time, we observe that both LAWA and EMA can significantly speed up training with respect to the considered baseline (Figure 1). If a target validation score is known in advance, weight averaging can effectively be employed to trigger early stopping and significantly reduce computational costs. We estimate a 12% reduction of GPU-hours for training AlgoPerf using LAWA on top of NadamW.

**Averaging horizon.** We further investigate the relationship between the optimal averaging window and training efficiency. Figure 3 reveals a consistent trend across workloads: whereas moderate weight averaging significantly reduces the number of steps required to reach the validation target, both excessively small and excessively large horizons can be suboptimal, resulting in minor speed-ups. Importantly, we find that LAWA is often highly stable and

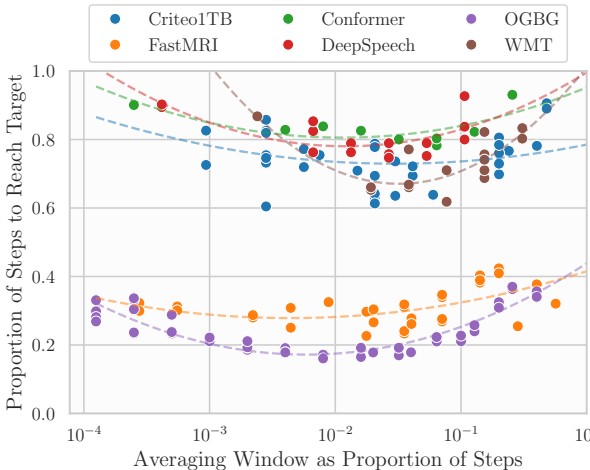

Figure 3: Impact of averaging horizon on training efficiency. Moderate weight averaging generally reduces the number of steps needed to reach the validation target. However, excessive averaging (rightmost region) or overly small averaging windows (leftmost region) may diminish gains or hinder progress. We use LAWA for this analysis, and define the averaging window proportion as $\frac{\nu \times L}{T}$ ($x$-axis), where $T$ denotes the total training budget, $\nu$ is the number of steps between consecutive checkpoints, and $L$ is the span of the averaging window. For each workload, we fit and display a second-order polynomial: $y = ax^2 + bx + c$.

forgiving with respect to the choice of its hyperparameters, considering that for each workload a broad range of horizons are effective. This robustness is particularly evident in OGBG and FastMRI, where performance remains relatively stable across a wide range of averaging windows. We attribute this to their long training horizons, as also noted in Kasimbeg et al. (2025). Furthermore, our analysis suggests a generalizable optimal averaging horizon across workloads: an averaging window around 1% of the total training budget consistently yields optimal results. We further explore the effect of LAWA and EMA hyperparameters on training speed-ups in Appendix B.

**Stability to hyperparameter tuning.** An important question is whether these gains are to be observed only at peak tuning of an optimizer, or if they hold in generic hyperparameter settings. We investigate this circumstance by variating the top learning rate of NadamW and averaging the correspondent checkpoints. We note that LAWA and EMA maintain their efficiency gains across all considered learning rates, reliably achieving the target score faster than the baseline optimizer, as shown in Figure 4. Averaging remains effective in improving efficiency, even when applied to a suboptimal optimization algorithm. This is encouraging, as it suggests potential benefits even in the absence of a strong baseline or when searching for one is too costly.

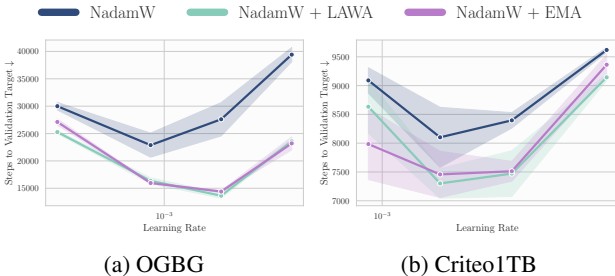

| (a) OGBG | (b) Criteo1TB |

Figure 4: Averaging weights of a suboptimal baseline. We train using NadamW, varying the top learning rate, and compare the number of steps needed to reach the validation target. We report mean and standard error across random seeds.

Similarly to Sanyal et al. (2023), we observe a slight advantage on OGBG with higher learning rates, but do not see a similar behavior for Criteo1TB; nevertheless on the latter WA remains effective at larger learning rates, in contrast to NadamW, which performance degrades faster. Interestingly, we notice that, as long as the baseline algorithm is able to reach the predefined target, weight averaging consistently speeds up convergence. If instead the learning rate is too big (or too small) and NadamW does not achieve the target in the maximum number of iterations, neither do the averaged checkpoints. This suggests that while averaging can significantly accelerate training, it has a limited impact on improving generalization—a point we explore further in the next section.

**Averaging Shampoo.** Finally, we explore whether a more sophisticated optimizer can benefit from weight averaging. We choose Distributed Shampoo (Shi et al., 2023) as the best scoring algorithm on the AlgoPerf inaugural competition (Kasimbeg et al., 2025), which provides a significant speed-up over NadamW. We report in Figure 5 the number of steps required to reach the validation targets when training an encoder-decoder Transformer on WMT, a ViT on ImageNet and a U-Net on FastMRI. We equip Distributed Shampoo with LAWA and EMA, and observe that weight averaging consistently enhances performance, even when applied to an already highly efficient optimizer like Shampoo, reaching the targets sooner and occasionally reducing the variability of the baseline optimizer. This emphasizes that the benefits of weight averaging extend beyond compensating for slower or less effective optimizers, and indicates that averaging offers inherent advantages, regardless of the adopted optimizer.

## 5. Improving Generalization

When a predefined target is not known *a priori*, or a computational budget is fixed and one does not need to stop training

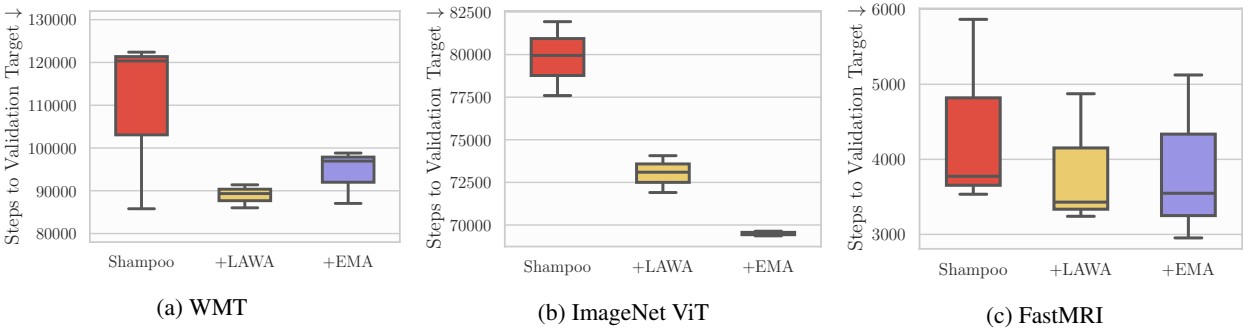

(a) WMT                  (b) ImageNet ViT                 (c) FastMRI

Figure 5: Weight averaging provides significant speed-ups also when applied on top of a more sophisticated optimizer like Distributed Shampoo. Averaging checkpoints through LAWA or EMA reduces the number of iterations required to reach the validation targets on the considered workloads.

Table 2: Validation performance across workloads. Both averaging schemes show minimal improvement over the baseline, always matching or slightly surpassing its performance. We report the mean and standard deviation across 3 seeds, for all workloads but Imagenet ViT, due to its high computational cost. We observe notable gains in the WMT and OGBG workloads. We do not find a consistent significant difference between the two averaging schemes, we noice that better hyperparameter configuration for LAWA and EMA are possible and may lead to larger improvmentes.

|  | CONFORMER WER ↓ | DEEPSPEECH WER ↓ | CRITEO1TB Loss ↓ | OGBG MAP ↑ | FASTMRI SSIM ↑ | WMT BLEU ↑ | ViT Acc ↑ |
|---|---|---|---|---|---|---|---|
| NADAMW | $0.08849_{\pm 0.01066}$ | $0.11894_{\pm 0.00614}$ | $0.1235_{\pm 0.00003}$ | $0.3012_{\pm 0.00118}$ | $0.72690_{\pm 0.00021}$ | $31.157_{\pm 0.06373}$ | $0.79048$ |
| +LAWA | $0.08845_{\pm 0.01067}$ | $\mathbf{0.11613}_{\pm 0.00136}$ | $0.1235_{\pm 0.00004}$ | $\mathbf{0.3129}_{\pm 0.00397}$ | $0.72705_{\pm 0.00020}$ | $31.429_{\pm 0.06088}$ | $0.79112$ |
| +EMA | $\mathbf{0.08841}_{\pm 0.01073}$ | $0.11739_{\pm 0.00352}$ | $0.1235_{\pm 0.00004}$ | $0.3088_{\pm 0.00253}$ | $\mathbf{0.72707}_{\pm 0.00020}$ | $\mathbf{31.457}_{\pm 0.08598}$ | $\mathbf{0.79232}$ |

early, weight averaging might be an appealing option to improve generalization. We investigate this setting (Option B in Section 3) by fixing a step budget and comparing the *best validation score* achieved during training by the baseline algorithm and by its averaged version.

**WA leads to moderate performance gains.** We report in Table 2 and Figure 8 the results of averaging in terms of generalization performance. We find that using LAWA or EMA on top of a learning rate schedule consistently improves over the baseline optimizer, achieving slightly better validation scores across all the considered workloads. We notably observe significant improvements on the WMT workload. As previously reported in Kaddour (2022) and Sanyal et al. (2023), in the early stage of training, averaging schemes provide considerably better performance than the underlying optimization algorithm, but this gap shrinks towards the end of training. We argue that this behavior is closely linked to the use of WA on top of a learning rate schedule, and that the benefits of averaging diminish later in training due to its similarity to learning rate annealing (Sandler et al., 2023). We explore this topic in more details in the following section.

**Stability to hyperparameter tuning.** In line with the previous section, we investigate how averaging affects gener-

alization performance across different hyperparameter configurations. This is a common scenario, that may occur when an optimal baseline is not available, or when it is too expensive to search for one. We variate the baseline learning rate and report the achieved validation score with and without averaging. We observe in Figure 6 that the averaged checkpoints closely track the performance of the baseline algorithms, sometimes providing marginal gains. Unlike Sanyal et al. (2023), we do not notice inherent benefits when training at higher learning rates.

## 6. Averaging as a Proxy for LR Decay

To achieve optimal performance on most Deep Learning tasks, it is common practice to anneal the learning rate during training (Defazio et al., 2024a). Despite its ubiquity, this practice introduces extra hyperparameters, complicates resuming from the latest checkpoint (Singh et al., 2025) and significantly increases training costs (Hägele et al., 2024). In contrast, averaging weights along the training trajectory intuitively reduces noise and might act as a proxy for learning rate decay (Sandler et al., 2023; Defazio et al., 2024b).

In this section, we investigate how WA affects performance under various cosine LR schedules, whether it can reliably proxy a short LR decay, and whether it can fully replace it.

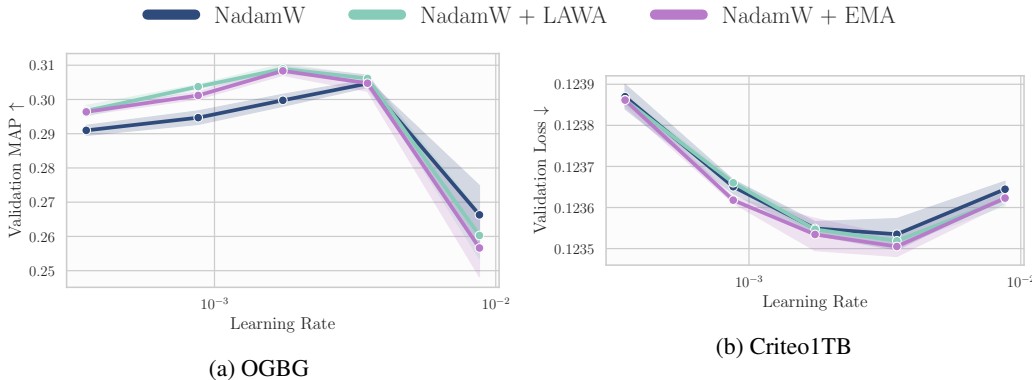

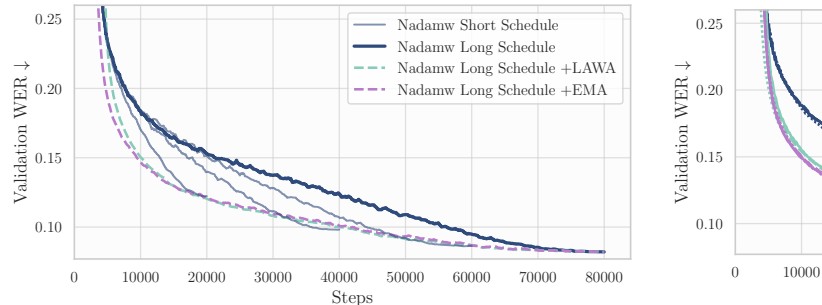

(a) OGBG

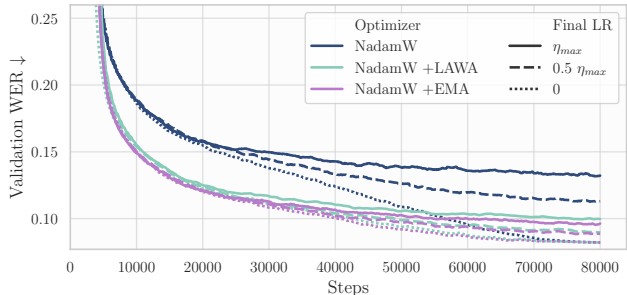

(b) Criteo1TB

Figure 6: Averaging a suboptimal baseline. We train with NadamW, varying the top learning rate, but still decaying it to zero, and compare the validation performance when averaging weights. We report mean and standard error across random seeds.

(a) Weight averaging of a long (annealed) training run performs similarly to training with shorter learning rate schedules. EMA and LAWA allow to materialize better-performing model for free withouth the need to lower the learning rate.

(b) Averaging checkpoints of training runs with different annealing strategies. The benefits of WA diminish when annealing the learning rate to zero. Here $\eta_{max} = 0.00175$, which correspond to the best configuration of NadamW on Librispeech Conformer.

Figure 7: Weight averaging on top of different learning rate schedules on Librispeech Conformer.

**Averaging vs LR scheduling.** In line with previous work, we compare checkpoint averaging over a long traning run against training with a *shorter learning rate schedule*. We systematically observe that the validation performance of averaged checkpoints closely tracks a training run with a faster learning rate decay (Figure 7a and Figure 12). This behavior, although pointed out in earlier studies, consistently emerges across the diverse model architectures and datasets considered for this analysis, and explains the considerable speed-ups observed in Section 4. Since checkpoint averaging acts as a proxy for a cooled-down learning rate, WA enables access to a better model during training, thus achieving the target score faster. We note that more sophisticated annealing strategies are possible (Hägele et al., 2024), and while the averaged model consistently approaches the Pareto frontier of loss versus training time (Portes et al., 2022), it may not always lie on it. Nevertheless, its simplicity and effectiveness make averaging a highly practical tool in large-scale training, providing access to a stronger model with minimal computational overhead.

Given the promising effect of averaging as an alternative to learning rate annealing, we examine its impact under different annealing strategies. We use a cosine learning rate schedule with a maximum value of $\eta_{max}$ and explore three variations: no annealing, annealing the learning rate to half of $\eta_{max}$, and annealing to zero (Figure 7b). Despite yielding significantly better validation scores during training across all learning rate schedules, the final validation performance of WA is strongly influenced by the annealing strategy. When little or no annealing is applied, WA provides substantial improvements; however, when the learning rate is fully annealed to zero, WA converges closely to the annealed model, suggesting that its benefits diminish as optimization naturally reaches a well-converged solution. This observation is consistent with our earlier findings, where averaging improves performance during training (Section 4) but provides minimal generalization gains when a learning rate schedule is used (Section 5), and further supports the previously discussed relation between averaging and learning rate scheduling.

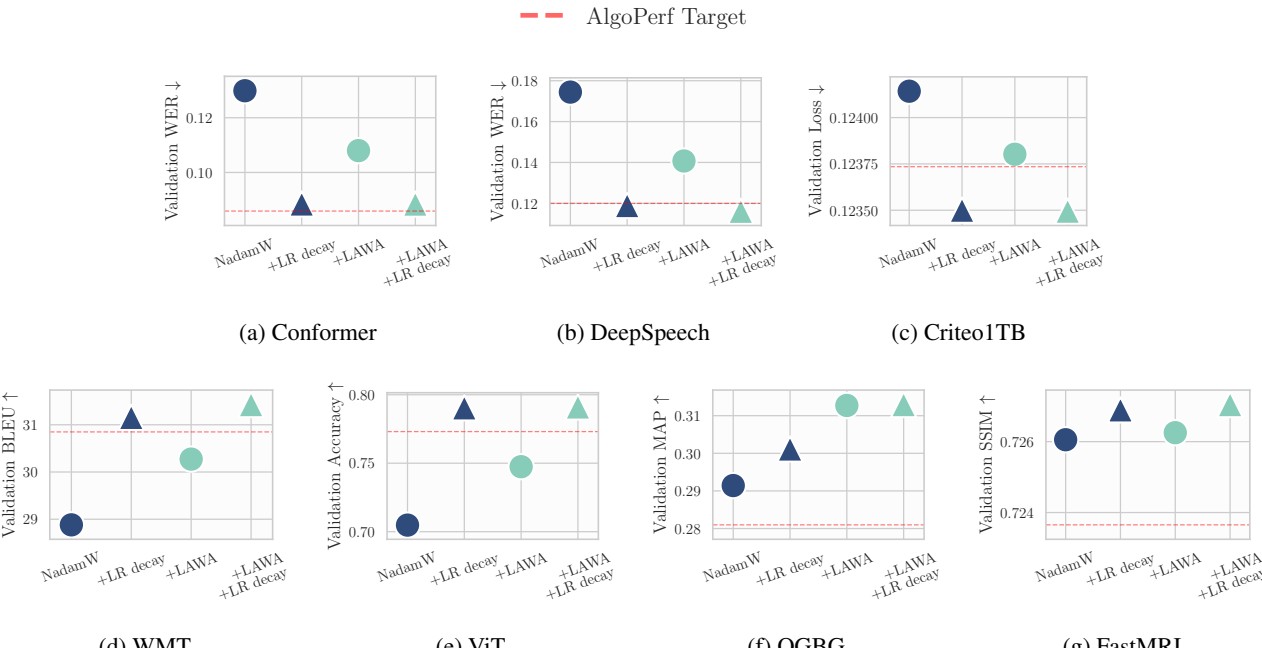

Figure 8: Generalization performance when training with NadamW, without (●) and with (▲) learning rate decay, versus training with NadamW +LAWA, without (●) and with (▲) learning rate decay. Despite significantly improving the performance of the baseline, weight averaging alone cannot generally match the validation performance of using a learning rate annealing schedule, and can rarely reach the target validation score. Instead, we observe that combining the two yields slightly better final validation scores throughout the considered workloads. We notice that OGBG and FastMRI are the only workloads where LAWA can provide a good substitute for LR decay, and attribute this behavior to the relatively easy target, very generous step budget, and high variability of these tasks, as also noted by Kasimbeg et al. (2025). We report the mean performance across 3 random seeds. The red dotted line displays the benchmark target scores.

**Can averaging replace LR decay on AlgoPerf?**  We thoroughly compare averaging with learning rate scheduling across the entire AlgoPerf suite, reporting our findings in Figure 8. Despite heavily tuning the averaging algorithms, we find that fully replacing the learning rate decay with averaging consistently yields inferior performances than annealing the learning rate, with the only exception of Citeo1TB and OGBG, where WA can achieve the target without lowering the learning rate. This result suggests that averaging schemes which do not use the average to compute optimization updates cannot fully replace a learning rate annealing, and more advanced techniques like the one introduced by Defazio et al. (2024b) are necessary to substitute LR decay. Nevertheless, we argue that these averaging schemes can still be used effectively and cheaply to materialize better-performing models during training, as shown in Section 4, with notable applications, including recent work by DeepSeekAI (2025) and Geiping et al. (2025). Additionally, applying them alongside a learning rate decay schedule can further enhance performance and improve generalization, all at minimal cost (Section 5).

## 7. Language Modeling

Although the AlgoPerf benchmark covers diverse workloads, including a machine translation task with an encoder–decoder transformer, contemporary language models predominantly adopt decoder-only architectures trained for causal language modeling (next-token prediction). We are therefore interested in understanding and benchmarking the potential impact of weight averaging on this widely adopted application. A previous work (Sanyal et al., 2023) has highlighted the benefits of early weight averaging on modern language models, showing how LAWA can produce stronger models early during training, reaching solid downstream tasks performance faster than the baseline trajectory. Unlike Sanyal et al. (2023), we show that an Exponential Moving Average of model parameters can successfully *match* the performance gains of LAWA when appropriately tuned. We further investigate the role of averaging hyperparameters, and comment on the connection between sampling frequency, rolling window length, and discount factor in LAWA and EMA.

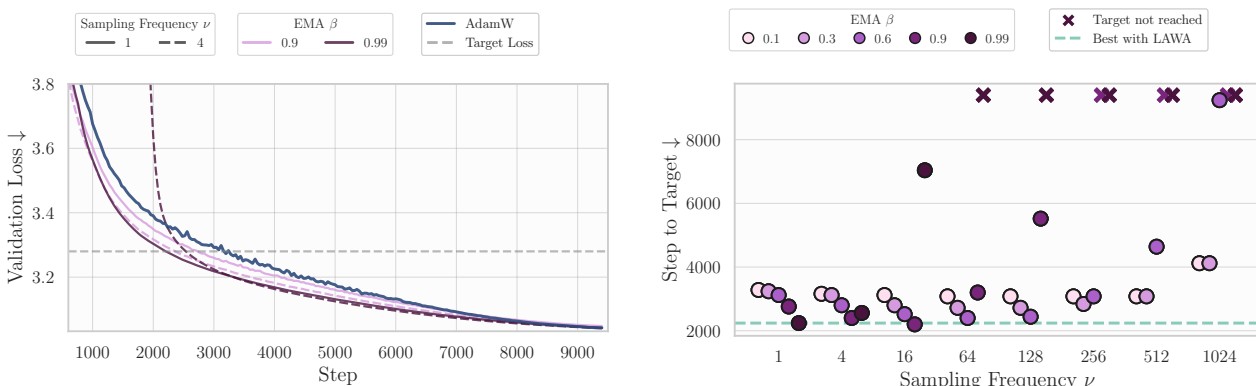

Figure 9: EMA speed-ups when training a 124M transformer on 5B FineWebEdu tokens. Many configurations outperform the baseline algorithm (AdamW, ≈ 3200 steps to target), but extreme values of $\beta$ or $\nu$ might hinder performance gains.

**Experimental Setting.** Since this analysis deviates from AlgoPerf, neither a well-defined target nor a strong baseline are *a priori* available. In an attempt to fairly evaluate WA on this task, we follow an approach similar to the recent nanogpt speed-runs (Jordan et al., 2024), and set the same *target validation loss* of 3.28 as our validation target score. We employ a similar 124M-parameters model, and the same GPT-2 tokenizer; training on 5B tokens from FineWebEdu (Penedo et al., 2024), and evaluating on a held-out split of 5M tokens. We use AdamW as the baseline optimizer, and perform a grid search over its hyperparameters to first build a strong reference, as described in Appendix D. We then apply LAWA and EMA on top of the fastest-to-target AdamW configuration and compare against it by means of step-to-target and final validation loss.

**Results.** Whereas AdamW reaches the target validation loss in about 3200 steps at best, both LAWA and EMA can shorten this time by roughly 30% (Figure 9; Figure 14a; Table 4). The magnitude of this speed-up clearly depends on the predefined target, and although the validation loss of the averaged checkpoints is dominated by AdamW's, the gap between the two shrinks as training proceeds—or more precisely, as the learning rate decreases. Indeed, we observe only minimal benefits in terms of final validation perplexity when comparing the final model obtained with AdamW with the one from LAWA and EMA (Figure 14b; Figure 13), similarly to Section 5. Interestingly, we find that EMA can accurately match LAWA performance when appropriately tuned. Figure 9 displays how EMA hyperparameters affect training speed-ups, showing how the optimal value of $\beta$ varies with the checkpoint sampling frequency ($\nu$). In our setting, the choice of $\nu = 1$, $\beta = 0.9$ from Sanyal et al. (2023) is indeed suboptimal, and a higher value of $\beta$ is needed when collecting checkpoint at every step. Similar conclusions hold when evaluating weight averaging by means of final generalization performance, as reported in Appendix D alongside LAWA results.

Although limited to small-scale experiments, this analysis confirms our earlier findings, and we hope it provides guidance when adopting weight averaging techniques in larger scale model training.

# 8. Limitations

We limit our analysis primarily to NadamW and AdamW as the baseline optimization algorithms. However, we believe that our conclusions are broadly applicable, and demonstrate how averaging can provide substantial benefits also on top of a higher-order optimizer like Shampoo. Nevertheless, it would be interesting to compare averaging schemes across a broader range of optimization algorithms, and against or on top of schedule-free methods (Defazio et al., 2024b).

# 9. Conclusions

Through a comprehensive empirical investigation, we study the effects of weight averaging on a challenging optimization benchmark (Dahl et al., 2023), consisting of several diverse Deep Learning architectures and datasets. We evaluate averaging methods by both the time required to reach a reasonable validation target and the best validation performance achieved, concluding that averaging offers substantial efficiency gains while achieving marginally better validation scores. Accessing the optimal model during training without prematurely decaying the learning rate remains an open fascinating challenge, and our work presents weight averaging as a step forward, moving models closer to the Pareto optimality of loss versus training time. We hope that this work further encourages the adoption of averaging techniques. Although such methods are not directly involved in optimizing updates, their significant improvements may dramatically reduce training time. In the field of efficient training, we believe this is a promising study with potential for practical applications (Shen et al., 2024).

## Acknowledgments

The authors would like to acknowledge the Hector Foundation for their support and funding, which made this research possible.

## Impact Statement

Our paper presents a thorough analysis of weight averaging for large-scale Machine Learning optimization, we acknowledge its potential impact in improving efficiency when training complex Deep Learning systems.

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

# A. Experimental Details

## A.1. Datasets and Architectures

We report further details about the workloads considered for the analysis and provide the correspondence references. We consider the following architectures and tasks:

- DLRMsmall model on the Criteo 1TB dataset (Naumov et al., 2019; Lab, 2014): a Deep Learning recommendation model optimized for large-scale industrial recommendation tasks.

- U-Net (Ronneberger et al., 2015) on the FastMRI dataset (Ronneberger et al., 2015; Zbontar et al., 2018): a popular architecture for medical image segmentation.

- ViT (Vision Transformer) on ImageNet-1k (Dosovitskiy et al., 2020; Deng et al., 2009): a transformer-based model for image classification.

- GNN model on OGBG dataset (Battaglia et al., 2018; Hu et al., 2020): a graph neural network for learning over graph-structured data, used to test performance on graph-based machine learning tasks.

- Transformer-big on WMT dataset (Vaswani et al., 2017; Bojar et al., 2017): a large-scale transformer model applied to machine translation.

- Conformer on LibriSpeech dataset (Gulati et al., 2020; Panayotov et al., 2015): a hybrid architecture combining CNNs and transformers, used for speech recognition tasks.

- DeepSpeech (Amodei et al., 2016) on LibriSpeech dataset (Panayotov et al., 2015): a recurrent model for speech-to-text, tested on a popular speech recognition dataset to assess its performance in transcribing audio.

These workloads span various domains, including recommendation systems, image and speech processing, machine translation, and graph-based learning. Each tests different model architectures under real-world data conditions.

## A.2. Computational Resources

We conducted experiments on 4xA100-SXM4-40GB and 4xH100-HBM3-80GB GPUs, occasionally resorting to 8xV100-32GB. We use Distributed Data Parallel to parallelize training across devices. Training is performed in full precision, enabling TF32-matmul for faster computations.

We make use of the original AlgoPerf repository[1], with minimal modifications, using the `PyTorch` (Paszke et al., 2017) implementation of the corresponding algorithms. For

the Distributed Shampoo implementation, we resort to the original submission to the AlgoPerf competition[2]. We select the optimal hyperparameters for NadamW and Shampoo on each workload as the ones providing faster convergence to the benchmark target in the first iteration of the AlgoPerf competition[3].

# B. LAWA and EMA Hyperparameters

Having demonstrated the benefits of averaging weights along the training trajectory, we now investigate the impact of the hyperparameters of the averaging scheme. Specifically, we explore the interaction between the update frequency $\nu$ and the averaging window $L$ for LAWA (Figure 10), as well as the relationship between $\nu$ and $\beta$ for EMA (Figure 11), and investigate how different choices of these hyperparameters influence the effectiveness of averaging in reducing the number of steps required to reach the benchmark validation targets. For this analysis we use NadamW as the underlying optimization algorithm, and the best performing hyperparameter from Dahl et al. (2023) for each workload.

In both cases, we observe a consistent trend: more frequent updates (lower $\nu$) benefit from longer memory (higher $L$ or $\beta$), and vice versa: sampling checkpoints further apart (higher $\nu$) works better with shorter memory buffers (lower $L$ or $\beta$). Several combinations of $\nu$ and $L$ or $\beta$ yield strong performance, suggesting that an optimal averaging horizon might exist. We also note that slower averages (high $L$ or $\beta$) are more sensitive to the choice of $\nu$, requiring more careful tuning, and that EMA is usually more sensitive to its hyperparameters. In practice, we find that values of $\nu$ in the range $64 - 128$ combined with $L = 20$ provide good performance across workloads for LAWA. For EMA, our analysis suggests that a value of $\nu$ between $64$ and $128$ with $\beta = 0.9$ is effective. We acknowledge that better configurations may exist, and that the optimal values may vary depending on the workload and on other hyperparameters, particularly on the LR schedule.

We further observe that when the baseline algorithm (NadamW) fails to reach the validation target—possibly due to poor initialization or an unfavorable data stream—applying LAWA or EMA typically does not recover performance and still fails to attain the target. This is evident, for instance, in the Librispeech Conformer and Librispeech Deepspeech workloads, where one of the random seeds leads the baseline to consistently fail, and both LAWA

---

[1] https://github.com/mlcommons/algorithmic-efficiency

[2] https://github.com/mlcommons/algorithms_results_v0.5/tree/main/AlgoPerf_Team_21/external_tuning/shampoo_submission

[3] https://github.com/mlcommons/algorithms_results_v0.5/tree/main/logs/algoperf_scoring_v05/external_tuning/AlgoPerf/prize_qualification_baseline

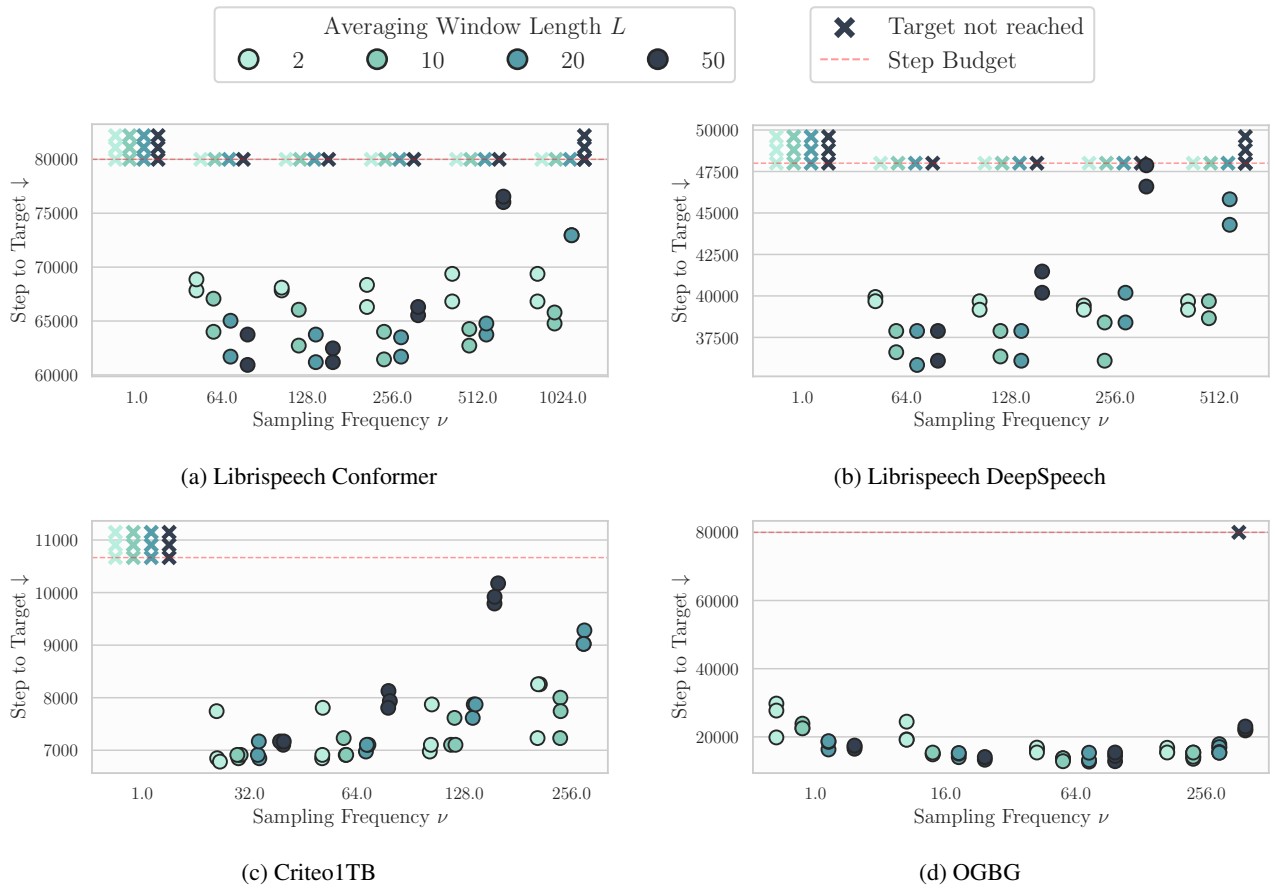

(a) Librispeech Conformer

(b) Librispeech DeepSpeech

(c) Criteo1TB

(d) OGBG

Figure 10: Performance of NadamW +LAWA across different combinations of sampling frequency $\nu$ and window length $L$ with each point representing a different random seed (3 seeds per configuration). An "x" mark indicates the target was not reached for that specific seed within the training horizon (red dotted line). We notice that small values of $\nu$ require a large value of $L$, and viceversa. On Criteo1TB, we exclude $\nu = 256$ and $L = 50$, as the first average would occur at step $256 \cdot 50 = 12800$ with this conifiguration, exceeding the training horizon (10666 steps).

and EMA exhibit the same behavior (indicated by the single persistent "x" mark across hyperparameter settings in Figure 10 and Figure 11). These observations reinforce the findings in Section 6: averaging can accelerate training by acting as a proxy for a shorter learning rate schedule, but it offers limited benefit when the baseline optimizer produces poor checkpoints and follows a suboptimal trajectory.

Finally, unlike Arpit et al. (2022), we observe that EMA with $\nu = 1$ performs poorly for $\beta \leq 0.99$. We find this behavior peculiar, and hypothetize that the worst performance of EMA compared to LAWA observed by Sanyal et al. (2023) might be due to updating the EMA buffer *at every step*, whereas for LAWA they use larger values of $\nu$, and further explore this in Section 7 and Appendix D. A similar result holds for LAWA, with the only exception of OGBG, where we however acknowledge a very long training horizon and a very slow baseline, as noted in Kasimbeg et al. (2025).

## C. Additional AlgoPerf Experiments

We report additional results for the analysis in Section 6, comparing averaging of a long annealed NadamW trajectory with shorter LR schedules across additional workloads. Figure 12 displays the decay schedule and validation performance during training for FastMRI, OGBG, and WMT. On FastMRI and OGBG, both LAWA and EMA closely match the performance of runs with a smaller budget. On OGBG, averaging generally provide better MAP scores throughout training but eventually regress to the overfitting baseline algorithm. On WMT, we observe that a short 25% schedule outperforms the averaging schemes at that point. Despite consistently approaching Pareto optimality of loss versus training time, LAWA and EMA cannot always achieve it. We argue that more sophisticated annealing strategies, possibly similar to Defazio et al. (2024b), may be required to access the optimal model at any time during training.

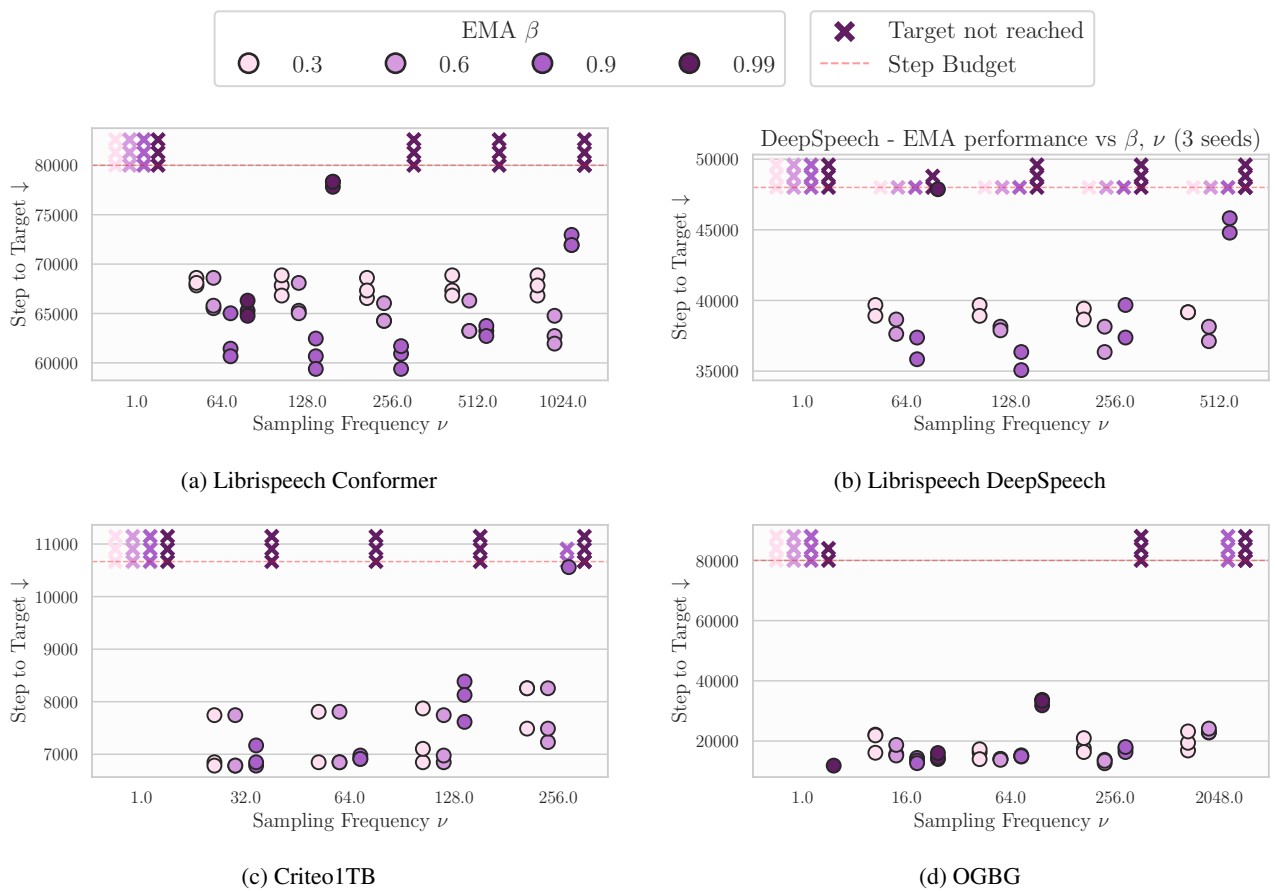

(a) Librispeech Conformer

(b) Librispeech DeepSpeech

(c) Criteo1TB

(d) OGBG

Figure 11: Performance of NadamW +EMA across different combinations of $\nu$ and $\beta$, each point represents a different random seed (3 seeds per configuration). Small values of $\nu$ generally work better with larger value of $\beta$, and vice versa.

## D. Additional Details on Language Modeling

We provide additional details and results for the Language Modeling analysis in Section 7.

We perform all experiments in `PyTorch` using `plainLM`[4] pretraining codebase and model implementations. The employed model is a decoder-only 124M transformer architecture, similar to GPT-2 (Radford et al., 2019; Karpathy, 2022), enhanced with Rotational Positional Embedding (Su et al., 2023), RMSNorm (Zhang & Sennrich, 2019), and SwiGLU (Shazeer, 2020). Models are trained for next-token prediction on 5B tokens from FineWebEdu (Penedo et al., 2024), using a batch size of 512, context length of 1024 tokens, and GPT-2 tokenizer. To derive a strong baseline algorithm, we perform a grid search over weight decay, gradient clipping and learning rate schedule hyperparameters, exploring both decay-to-zero and $10\%$ decay, as reported in Table 3. Having identified an optimal baseline across these values, we average on top of it using different LAWA

---

[4] https://github.com/Niccolo-Ajroldi/plainLM

and EMA configurations. We then compare these configurations by means of (A) step-to-validation-target and (B) minimal validation loss achieved throughout training, akin to Section 4 and Section 5 respectively.

Table 3: AdamW Hyperparameter Search Space. Final LR is reported as a fraction of top LR and warmup as a fraction of the step budget. We highlight in **blue** the fastest-to-target configuration after the grid search.

| Hyperparameter | Values |
|---|---|
| Top LR | 0.0003, 0.001, 0.003, **0.01**, 0.03 |
| Final LR | 0, **0.1** |
| Warmup | 0.01, **0.1** |
| Weight decay | 0.01, **0.1** |
| Gradient norm clipping | 0.01, **0.1** |

Results for optimal WA configurations are compared to the AdamW baseline in Table 4, and the interplay between $\nu$, $L$, and $\beta$ is explored in Figure 9, Figure 13, and Figure 14.

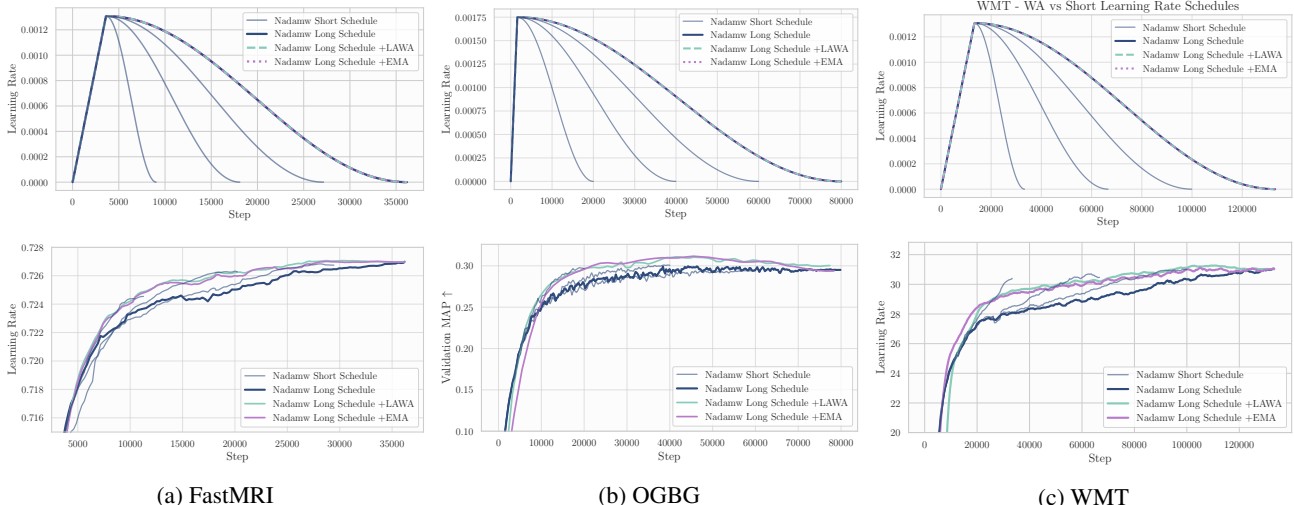

(a) FastMRI        (b) OGBG        (c) WMT

Figure 12: *Top*: Learning rate schedules for different workloads. *Bottom*: Corresponding performance during training. Weight averaging of a long annealed training trajectory approaches the performance of runs with shorter decay schedules.

Table 4: Language Modeling speed-ups and validation performance. WA provides substantial speed-ups, but minimal generalization gains over the last checkpoint. Results report median and standard deviation over 3 seeds. The missing standard deviation for the step-to-target is an artifact of the evaluation frequency, which we set to 40 steps, limiting the resolution of the correspondent metric, and resulting in a standard deviation of zero.

|        | Step-to-target | Best Validation Loss |
|--------|----------------|----------------------|
| AdamW  | 3200           | $3.0425_{\pm 0.003630}$ |
| +LAWA  | 2240           | $3.0421_{\pm 0.000002}$ |
| +EMA   | 2200           | $3.0413_{\pm 0.000002}$ |

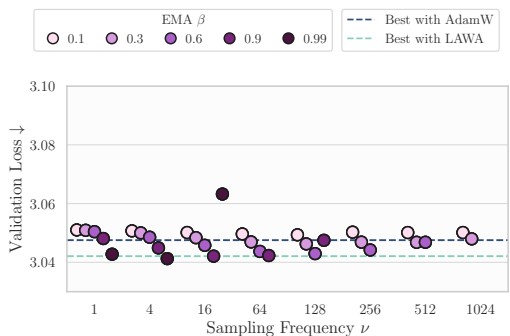

Figure 13: Language Modeling final validation loss of AdamW +EMA across different combinations of $\nu$ and $\beta$.

## E. Averaging Algorithms

We perform most of the experiments using offline versions of LAWA and EMA. This involves training with a baseline algorithm (NadamW, AdamW or Distributed Shampoo), frequently saving checkpoints to disk. This allows us to then average and evaluate the model weights, exploring different averaging windows and hyperparameters at a minimal cost, without having to retrain the model from scratch.

In practice, however, it might be beneficial to use online averaging schemes. As demonstrated throughout this work, averaging can operate as a proxy for a shorter learning rate schedule, providing significantly better performance than the currently available model. Since this aspect has interesting implications for large-scale efficient training, it is worth discussing the possible computational overhead introduced by averaging schemes. For the online versions of EMA or LAWA, we offload the averaging buffer to CPU.

We implement the online schemes naively in `PyTorch`, defaulting to blocking communication, but we note that it is possible to asynchronously offload the model weights to the CPU, using non-blocking communication to update an EMA or a moving average of model weights. We acknowledge a first adoption of this approach in DeepSeekAI (2025), and hope that this work encourages a wider usage of it.

Finally, we note that the poor performance of LAWA and EMA in the inaugural AlgoPerf competition (Kasimbeg et al., 2025) stemmed from the API's lack of support for switching between model parameters and averaging buffers at evaluation time, resulting in inefficient CPU-GPU transfers on bandwidth-limited hardware. Additionally, AlgoPerf does not support asynchronous transfers, further penalizing such submissions. However, as noted earlier, more efficient implementations are possible in real-world applications, where weight averaging can be used to seamlessly materialize better performing models. We report a `PyTorch`-style implementation of LAWA and EMA in Algorithms 1 and 2.

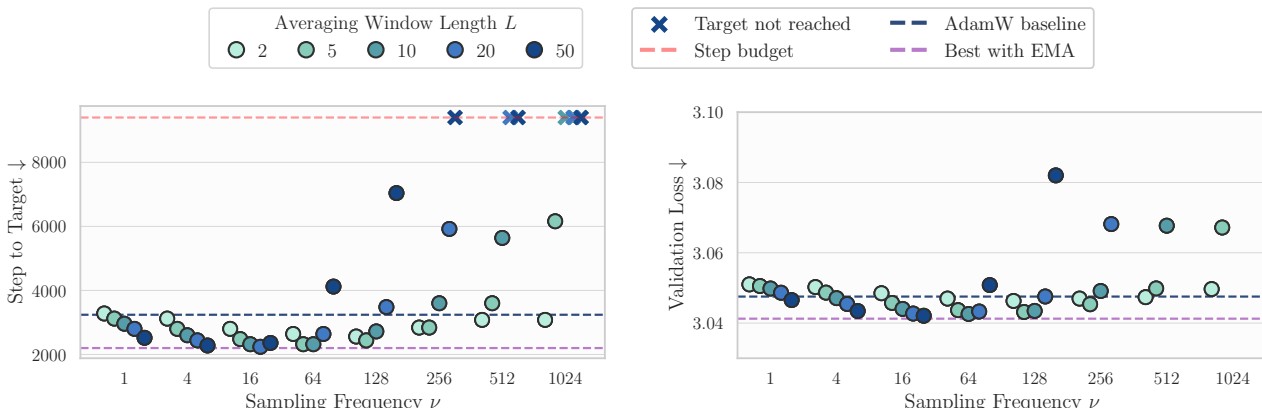

(a) Training speed-up. Training runs that do not reach the target validation loss by the end of trainng are marked with an "x".

(b) Final validation loss. Training runs that achieve a minimum validation loss $\geq 3.10$, are omitted.

Figure 14: Language Modeling performance of AdamW +LAWA across different combinations of $\nu$ and $L$. Small values of $\nu$ pair best with larger $L$, and vice versa. Beyond a *critical sampling frequency* $\nu$, weight averaging gains drop drastically, both in training acceleration and in final validation performance. We observe similar performance between LAWA and EMA (Figure 13, Figure 9), with both methods achieving comparable speed-ups and loss trajectories across most $\nu$ values; minor differences emerge only at extreme hyperparameter settings. These results suggest that, provided $\nu$ and $L$ (or $\beta$) are correctly initialized, either LAWA or EMA can be chosen without significant loss of efficacy.

---

**Algorithm 1** Latest Weight Averaging ( LAWA )

---

  Initialize LAWA hyperparameters $\nu$, $L$
  Initialize circular queue $Q$ of length $L$
  **for** $t$, batch in enumerate(trainloader) **do**
    Forward, backward and optimization step
    **if** $t \bmod \nu = 0$ **then**
      **if** $|Q| < L$ **then**
        Add current parameters $\theta_t$ to $Q$
      **else**
        Remove oldest element from $Q$, add current parameters $\theta_t$
      **end if**
    **end if**
    **if** is eval time **then**
      $\bar{\theta}_Q \leftarrow \frac{1}{L} \sum_{\theta \in Q} \theta$
      Eval $\bar{\theta}_Q$ on held-out data
    **end if**
  **end for**

---

**Algorithm 2** Exponential Moving Averaging ( EMA )

---

  Initialize EMA hyperparameters $\nu$, $\beta$
  Initialize EMA parameters $\theta_{\text{EMA}} = \theta_0$
  **for** $t$, batch in enumerate(trainloader) **do**
    Forward, backward and optimization step
    **if** $t \bmod \nu = 0$ **then**
      $\theta_{\text{EMA}} \leftarrow \beta\theta_{\text{EMA}} + (1-\beta)\theta_t$
    **end if**
    **if** is eval time **then**
      Eval $\theta_{\text{EMA}}$ on held-out data
    **end if**
  **end for**

---

