# OpenReview forum: "When, Where and Why to Average Weights?"
_ICML.cc/2025/Conference — ICML 2025 poster_

### Official Review · Reviewer_fFcZ · 2025-02-19

**Overall Recommendation:** 4

**Summary:**

The paper studies weight averaging, a "trick" rooted from the old Polyak averaging that never disappoints practitioners from fitting logistic regression to training modern LLMs. This paper provides a relatively large-scale study on AlgoPerf, a benchmark suite for optimizer, aiming at understanding: 1. Whether you should always use weight averaging; 2. How you set the weight averaging time window; 3. How you should use weight averaging together with learning rate decay.
The paper shows that: Using weight averaging always gives you faster convergence (i.e. fewer iterations and wall clock time) to a targeted validation loss and using a window size of 1% of the total training steps seems to be consistently the optimal option. Lastly the paper shows that although weight averaging has some connection with learning rate scheduling and can replace scheduling in some cases (e.g. The Road Less Scheduled, Defazio 2024), weight averaging still shows better performance when combined with standard learning rate scheduling.

Overall, the problem studied is important and timely, the paper is mostly clearly written but I find some parts lacking details and need more empirical evidence.

# Post rebuttal: The author agreed to include more discussion for the choice of WA parameters as well as other modifications I suggested. Based on that, I have increased my score from 3 to 4.

**Claims And Evidence:**

The empirical results for LAWA and EMA being helpful is clear and solid: Given a known optimal hyperparameter setting, applying WA always boosts the performance: Fig.1 and 2.

The results for other claims are less thoroughly evaluated, e.g.
- the combination with Shampoo is only evaluated on one dataset
- Comparision of WA + long decay schedule with short decay schedule without WA (Fig. 7a) seems to be only based on one model. Plus the model used is not stated in the caption.

For the recommendation of averaging window size, I think there are a couple of factors that could be important but not taken into account:
- Batch size (see e.g. https://arxiv.org/abs/2307.13813)
- Weight decay
- Learning rate warmup

**Essential References Not Discussed:**

Not any I am aware of.

**Experimental Designs Or Analyses:**

The experiment design aligned with the claims the paper is trying to make.

The most crucial hyperparmaeter: Learning rate, is swiped (Fig. 4).

However, the type of learning rate scheduling considered is limited only to cosine schedule, some more recent advances such as WSD or learning rate schedule with warmup (although I guess the observations will still hold true).

**Methods And Evaluation Criteria:**

AlgoPerf used in the paper seems to be a sensible evaluation benchmark.

**Other Comments Or Suggestions:**

I think Appendix A.2 could be moved into the main text since there are still sufficient space in the main text, and this seems to be pretty critical implementation details since it is highly related to the computation of the wall clock time and practical efficiency of WA.

**Other Strengths And Weaknesses:**

It would be much better if the weight averaging strategies considered can come with some formulas to help readers understand better, the current version describes EMA and LAWA with only languages, which I don't find very straightforward.

**Questions For Authors:**

In page 4 bottom, it is mentioned that "whereas a short one may be suboptimal or result in a serious computational overhead.", can the authors elaborate more on the computational overhead part? I don't see why it is the case if you are using, e.g. EMA.

**Relation To Broader Scientific Literature:**

The idea and the observations are mostly known, and as the authors admit in the submission, many observations confirm the conclusion in past literature. However, I am not aware of any existing papers that actually perform such large-scale studies. So I think the paper could be helpful for the community as a source of empirical evidence.

However, on the other hand, I am not sure if there are any new "insights" this submission provides to the community. The observation I find most interesting is the one in Fig.7a, where the authors show that a long run with weight-averaging, at the middle of training, gives you a performance similar to a full shorter run. I think this observation is relatively novel (at least for me) and could potentially help practitioners perform e.g. continuous pre-training, or tuning the training length, easier.

**Theoretical Claims:**

There are no theoretical claims in the submission.

---

> ### Author Rebuttal · Authors · 2025-03-31
>
> Thank you for the thoughtful comments. We appreciate your recognition of the value of our study! We address specific points below.
> > The results for other claims are less thoroughly evaluated, e.g. the combination with Shampoo is only evaluated on one dataset. Comparision of WA with LR schedules (Fig 7) seems to be only based on one model.
>
> We acknowledge that the mentioned experiments were missing some data, and we are happy to share **additional results**:
> - We performed [additional experiments on Shampoo](https://anonymous.4open.science/r/ICMLE1C2/Shampoo.pdf), considering **more workloads** and including **EMA**. Shampoo consistently hits the target faster across tasks when equipped with LAWA or EMA.
> - [WA vs Short LR schedule](https://anonymous.4open.science/r/ICMLE1C2/Short_LR.pdf) We report results on additional **workloads** and on **EMA**. LAWA and EMA yield better performance throughout training, closely matching the results from shorter schedules.
> - [We also included EMA](https://anonymous.4open.science/r/ICMLE1C2/Annealing_EMA.png) in the previous analysis on Librispeech Conformer, and find results to be consistent with the one on LAWA. When little or no annealing is applied, EMA provides substantial improvements on the final validation performance; however, when the LR is fully annealed to zero, EMA converges closely to the annealed model.
> - [LR sweep on EMA](https://anonymous.4open.science/r/ICMLE1C2/LR_Sweep.pdf): We add results on EMA when testing WA at different learning rates, observing that EMA efficiency is preserved across LR's.
>
> > For the recommendation of averaging window size, I think there are a couple of factors that could be important: batch size, weight decay, learning rate warmup.
>
> We agree that **batch size** and **weight decay** might affect the optimal averaging configuration of LAWA and EMA. We appreciate you highlighting this, and we will include this clarification in the revised version of the paper. However, we would like to emphasize that:
>
> 1. Due to the high computational cost of AlgoPerf and limited resources, we were unable to sweep all hyperparameters and instead focused on the best-performing combinations for each workload.
> 2. These configurations account for a variety of hyperparameters, with batch size, weight decay, and learning rate warmup varying across workloads. As such, our recipe is relatively robust to these choices, though some variability may still occur.
> 3. Finally, as suggested by Reviewer #egdu, [we included additional results on the effect of $\beta$ and $\nu$](https://anonymous.4open.science/r/ICMLE1C2/EMA.pdf) on EMA performance. We refer to the answer to Reviewer #egdu for a discussion on these additional experiments.
>
> > The type of learning rate scheduling considered is limited only to cosine schedule, more recent advances such as WSD or learning rate schedule with warmup would be interesting.
>
> We would like to clarify that the employed schedule is Cosine Decay **with warmup**. We apologize if this was not clear and will ensure it is explicitly stated in the revision. Regarding recent advances like **WSD**, we find this to be a very interesting question! We would be particularly interested in whether averaging can accelerate the decay phase of WSD, potentially **reducing the computational cost of cooldowns**. We believe this is a promising area for future work, and hope that our study inspires further research on this topic.
>
> > It would be much better if the averaging strategies considered can come with some formulas to help readers understand better.
>
> Thank you for pointing this out! We had included the algorithms in an earlier version of the paper but excluded them for the submission. [We display them at this URL](https://anonymous.4open.science/r/ICMLE1C2/Algorithms.png) and will include them in the Appendix of the revised manuscript.
>
> > Can the authors elaborate more on the computational overhead part? I don't see why it is the case
>
> You are correct in noting that averaging does not incur additional overhead when implemented efficiently. In the quoted sentence, we were referring to a naive implementation of WA, where the average is offloaded to CPU memory and then transferred back to CUDA memory before evaluating the model. We appreciate the question and will revise the sentence to make this distinction clearer. For more details on this topic and data about the overhead, please refer to our response to Reviewer #egdu and this analysis.
>
> **To conclude**, we sincerely thank you for your review and thoughtful questions. The discussion has been valuable, and we believe that addressing these points and providing additional evidence has strengthened our work by broadening the scope of our analysis and better supporting our claims. We hope to have clarified your concerns, and we would greatly appreciate if these improvements could be reflected in a higher evaluation. We are happy to address any further questions or doubts.

---

> > ### Comment · Reviewer_fFcZ · 2025-04-02
> >
> > I would like to thank the author for the detailed response! My rating stays unchanged: I think this is a good paper that contributes **a lot of useful evidence** and **great value** to the community.
> >
> > However, I have another question for the authors: Imagine I am a practitioner trying to train a model, now after I read the paper, I know that I should use learning rate scheduling with weight averaging, but **how should I set the WA parameters**? Should I treat them as hyperparameters, use a heuristic value (i.e. arbitrary value should be fine), a heurstic value in a range, etc. I would love to see the authors provide a more detailed guideline / or plan for revision on "when, where and how" to perform weight averaging.
> >
> > Note that I am also totally fine if the authors admit that this is an open question and there is no good answer to it. If that is the case, then I believe it would be better if the authors stated that in a "summarization" section, otherwise it could be disappointing for the readers who believe the paper provided a "complete recipe" but ended up failing to find the desired answer.
> >
> > With this point addressed, I will increase my score.

---

> > > ### Author Response · Authors · 2025-04-07
> > >
> > > Thank you for the response!
> > >
> > > We indeed have some data on the effect of **WA hyperparameters**, and agree that discussing these results in the paper would be useful for the community. We partially address this question in Figure 3, but are happy to share additional results here.
> > >
> > > We study the interaction between the update frequency $\nu$ and the averaging window $L$ for LAWA at the following link: https://anonymous.4open.science/r/ICMLE1C2/LAWA.pdf, and between $\nu$ and the EMA discount factor $\beta$ here: https://anonymous.4open.science/r/ICMLE1C2/EMA.pdf
> > >
> > > In both cases, we observe a consistent trend: more frequent updates (lower $\nu$) benefit from longer memory (higher $L$ or $\beta$), and vice versa, sampling checkpoints further apart (higher $\nu$) works better with shorter memory buffers (lower $L$ or $\beta$). Several combinations of $\nu$ and $L$ or $\beta$ yield strong performance, suggesting that an optimal averaging horizon might exist. We also note that slower averages (high $L$ or $\beta$) are more sensitive to the choice of $\nu$, requiring more careful tuning, and that EMA is usually more sensitive to its hyperparameters.
> > >
> > > In practice, we find that values of $\nu$ in the range 64–128 combined with $L=20$ provide good performance across workloads for LAWA. For EMA, our analysis suggests that a value of $\nu$ between 64 and 128 with $\beta=0.9$ is effective. For Criteo1TB, which has a much shorter training horizon, a smaller value of $\nu=32$ proves more beneficial. We acknowledge that better configurations may exist, and that the optimal values may vary depending on the workload and on other hyperparameters.
> > >
> > > Finally, we observe that EMA with $\nu=1$ performs poorly for $\beta \leq 0.99$. We find this behavior peculiar, and hypothetize that the worst performance of EMA compared to LAWA observed by Sanyal et al. (2023) might be due to *updating the EMA buffer at every step*, whereas for LAWA they use larger values of $\nu$. We plan to further explore this in future work. A similar result holds for LAWA, with the only exception of OGBG, where we however acknowledge a very long training horizon and a very slow baseline, as noted in Kasimbeg et al. (2025).
> > >
> > > We thank you for your insightful comments and will include these additional details in the revised manuscript.
> > >
> > > ---
> > > **References**
> > >
> > > Sanyal, S., Neerkaje, A., Kaddour, J., Kumar, A., & Sanghavi, S. (2023). Early weight averaging meets high learning rates for LLM pre-training.
> > >
> > > Kasimbeg, P., et al. (2025). Accelerating neural network training: An analysis of the AlgoPerf competition.

---

### Official Review · Reviewer_egdu · 2025-03-06

**Overall Recommendation:** 3

**Summary:**

This submission presents the experimental findings and evaluation of two weight-averaging (WA) techniques, LAWA and EMA, on the AlgoPerf optimization benchmark, on seven tasks. They find strong training speed improvements (15% reduction in GPU-Hours), consistent with varying hyperparameters (learning rate and averaging window) with second-order optimizers. They also show a small improvement in generalization provided by WA. Finally, they provide experiments linking WA to learning rate annealing, showing that the benefits of WA diminish when annealing the LR to zero.

[Score raised from 2 to 3]

**Claims And Evidence:**

The authors claim that their experiments show that WA methods speed-up training improve generalization, and that WA is a proxy for shorter learning rate decay, but can not replace it completely. I think that these claims are relatively well confirmed by the experiments presented, with a few caveats:

I disagree with the claim that "show that coupling WA with learning rate annealing yields optimal results" (l49), since they later confirm that learning rate annealing is particularly the case where the benefits of WA disappear (Fig. 7b).

Similarly, the generalization improvement provided by WA is very limited (but consistent).

Finally, see my main Question regarding speeding-up considering the results of the AlgoPerf competition (Anonymous, 2025 in the submission), which show WA methods as slower than the baseline.

**Essential References Not Discussed:**

Most essential references have been discussed. I find the Model Soups part lacking more citations. The links between WA methods and self-supervised approaches could also be discussed, as well as more modern approaches using WA methods.

**Experimental Designs Or Analyses:**

The experiments done by the authors seem valid and consistent with previous findings. I voiced the concerns I had in other sections (relative lack of variety in experiments, inconsistent results in the AlgoPerf results..)

**Methods And Evaluation Criteria:**

The authors use 7 (of the 8) benchmarks of AlgoPerf to study the effectiveness of LAWA and EMA for deep learning optimization. This provides a thorough and comprehensive optimization landscape to study their claims. They also provide experiments regarding learning rate schedule, averaging window size and an experiment on a second-order optimizer.

I think only studying two WA methods is a bit limited, as this would help better understand and choose WA options through extensive benchmarking, such as the effect of overfit-aware sampling and dense sampling. I'm also surprised the dependence on the EMA coefficient and update frequency was not studied like the window size, in particular when the authors claim (l189) to explore the interaction of the frequency variables with others.

I am also surprised that many of the experiments (Figures 3, 4, 5, 6, 7, 8) only consider LAWA and neither EMA. This feels like an important oversight to justify the general and consistent effectiveness of WA methods to only consider one of the two approaches.

**Other Comments Or Suggestions:**

I'm a bit unsure if a second-order polynomial is an appropriate fit for the results in Figure 3, in particular for the upper results (Criteo, Conformer, DeepSpeech etc). Can the authors also provide the values for runs that result in a proportion over 1?

Please provide the value of $\eta_{\text{max}}$ (line 291).

Can the authors please detail the link between WA and learning rate decay that they find in Theorem 5.3. in Garrigos & Gower ?


l67/68: additional ","

l316: "Citeo1TBB"

Fig 7b: "diminsh"

**Other Strengths And Weaknesses:**

This paper is well-written and clear, showing the effectiveness of WA clearly.

I find these results interesting, but I think that all results (except the combination with Shampoo, and the experiment in Fig 7b and 8) were already shown theoretically or experimentally by previous works: the speed-up and generalization improvement, and the main empirical link between WA and learning rate decay. I feel that this work either warrants further experiments with various WA approaches, or at least varying the hyperparameters missing (EMA coefficient our update frequency); or some more novel insights to justify an acceptance, but I can be convinced otherwise. As said by the authors (l361), studying a more broad range of optimizers could be interesting.

**Questions For Authors:**

I have an important caveat on the results of the authors. Following Anonymous, 2025 (the results of the AlgoPerf competition cited by the authors); LAWA and EMA perform worse than the NadamW baseline (Fig 1. a/b, Table 1). I'm having a hard time making sense of the authors' results in this paper and the ones reported in the AlgoPerf competition, which seems to show that averaging weights only resulted in slower training. One reason I can find for this discrepancy is that the authors' report mainly the number of optimization steps and not the training time, which matters in the benchmark competition. However, this is not the case anywhere, and many claims in the submission are related to GPU-Hours training time (Tab 1 for instance). What is the reason for this discrepancy? Could the authors give the additional time that the averaging steps add to the standard training time, to better understand this possible slowdown? What throughput difference do the offline and online versions have?

Why did the authors not also report the performance for the ImageNet image classification task with a ResNet, the last benchmark in the AlgoPerf suite?

It is not clear what is the benchmark considered in Figure 7.

The difference in results between LAWA and EMA is not really analyzed in the article if I'm not mistaken.

What is the answer to the title of this paper? The answer seems to be "always use WA, as a way to better do learning rate decay" to me while reading.

**Relation To Broader Scientific Literature:**

This work is an experimental benchmarking of the effectiveness of WA methods for deep learning optimization. It is related to the WA and optimization literature, mainly confirming pre-existing findings on large optimization tasks.

**Theoretical Claims:**

There are no theoretical claims in the article, but links with previous theoretical claims which are confirmed in this work are presented.

---

> ### Author Rebuttal · Authors · 2025-03-31
>
> Thank you for the thorough review! We appreciate the detailed feedback.
> > I think only studying two WA methods is a bit limited.
>
> We agree that broader benchmarking would be valuable and hope this works encourages further study at scale. We focus on LAWA and EMA due to their success and adoption. Influential works such as Szegedy et al. (2016), Vaswani et al. (2017), and Merity et al. (2017) used **one of these two averaging flavors**. We briefly tested Polyak averaging, but found it ineffective, consistent with Defazio et al. (2024), and omitted it. Finally, AlgoPerf's high computational cost limits extensive experimentation, so we prioritized exploring LAWA and EMA in depth.
> > I'm surprised the dependence on the EMA coefficient and update frequency was not studied.
>
> Thank you for mentioning this! We recognize its importance and are happy to share additional results: https://anonymous.4open.science/r/ICMLE1C2/EMA.pdf
> 1. Small values of $\nu$ require a large $\beta$, and viceversa.
> 2. Update frequency around 0.003 with $\beta=0.9$ work well across tasks.
> 3. $\nu=1$ (first group in each plot) performs poorly for $\beta\leq0.95$
>
> > Many experiments only consider LAWA and not EMA.
>
> We acknowledge this and would like to share **additional experiments**:
> - [Shampoo](https://anonymous.4open.science/r/ICMLE1C2/Shampoo.pdf) We include **more workloads** and **EMA**. Shampoo consistently hits the target faster across tasks when equipped with LAWA or EMA.
> - [LR schedule](https://anonymous.4open.science/r/ICMLE1C2/Short_LR.pdf). Results on more **workloads** and **EMA** confirm that WA closely matches a short LR schedule.
> - We test [EMA with different annealing](https://anonymous.4open.science/r/ICMLE1C2/Annealing_EMA.png), and find consistent results with LAWA.
> - EMA efficiency is preserved [across different LR's](https://anonymous.4open.science/r/ICMLE1C2/LR_Sweep.pdf), similar to LAWA.
>
> > Can the authors detail the link between WA and LR decay that they find in Thm 5.3. in Garrigos & Gower?
>
> Thm 5.3 provides convergence results for SGD under decreasing stepsizes, requiring an averaging scheme adapted to the stepsize schedule. A further connection to averaging stems from comparing this result to Lemma 7.3 where the gradient is computed at the running average of the iterates (see also Thm 7.4).
> >  Following Anonymous, 2025, WA perform worse than NadamW. What is the reason for this discrepancy? Could the authors give the additional time that averaging adds to training?
>
> This is a subtle point, thank you for raising it.
> 1) The AlgoPerf competition API did not allow switching model parameters to the WA buffer before evaluation, resulting to a suboptimal WA implementation. [This plot shows the](https://anonymous.4open.science/r/ICMLE1C2/Overhead.pdf) **overhead** of EMA. On Criteo it's 10x slower.
> 2) AlgoPerf does not allow **asynchronous** CPU-GPU transfers and has limited VRAM, requiring WA buffers to be stored in CPU memory and transferred frequently over bandwidth-constrained hardware.
>
> However, **WA can be implemented efficiently** using asynchronous non-blocking transfers, as shown by Deepseek AI et al. (2024). We used offline averaging to reduce computational costs, but an efficient implementation would have negligible overhead.
> > The difference between LAWA and EMA is not really analyzed in the article.
>
> We do not find significant difference between the two (Fig 2). Moreover, our goal was not to find _the best averaging scheme_, so we omitted a thorough comparison. However, we agree this could be better discussed and will update the paper.
> > What is the answer to the title of the paper?
>
> WA should always be used with a LR schedule to produce better models **for free** during training. How to access *the optimal* model at any time during training remains an open fascinating question. Our study presents WA as a step in this direction, moving the model closer to Pareto optimality of loss vs training time. This is precisely the reason behind the observed efficiency gains. We will clarify this point in the article.
>
> ### Additional comments
> - We exclude ResNet because neither NadamW nor Shampoo reach the target (Kasimbeg et al, 2025), leaving no baseline for comparison.
> - Fig 7 is Librispeech Conformer, $\eta_{max}=0.00175$, which is the best configuration from Dahl et al. (2023).
> - Fig 3: the y-axis shows the proportion of step wrt training horizon. Runs with $y>1$ do not reach the target and are hence excluded
>
> Finally, while some aspects of our work were explored in prior studies, our contribution expands their scale and scope. We believe our paper helps unify existing findings, evaluating them across tasks, and providing thoroughly validated claims.
>
> ### Conclusion
> We appreciate the insightful questions. We think that these additional analyses will significantly enhance our study and reinforce its contributions. We hope to have addressed your concerns, and if so, we would appreciate your endorsement for acceptance.

---

> > ### Comment · Reviewer_egdu · 2025-04-06
> >
> > **2 methods** Thank you for the answer. I agree that these methods are influential, but still feel that it warrants more justification in the final version to remove the need to focus on other approaches, such as for Polyak averaging as provided.
> >
> > **New experiments** Thank you for the additional experiments. I feel these help provide a strong and consistent overview of the effectiveness of averaging methods.
> >
> > **AlgoPerf times** These explanations help understand the underperformance of WA methods on AlgoPerf, and would also be a welcome addition to the final version of the paper. However, why is the discrepancy so much more important for the Criteo task compared to the others?
> >
> > **LAWA/EMA** The additional experiments provided help the comparison between the two methods. But as said by the authors, additional comments comparing the two approaches will be welcome.
> >
> > I thank the authors for their detailed and clarifying rebuttal, which strengthened their manuscript. Several of the points that have been answered still require a modification to the final manuscript, but I am raising my score to 3, asking now for acceptance.

---

> > > ### Author Response · Authors · 2025-04-07
> > >
> > > Thank you for the comment and feedback!
> > >
> > > The significant overhead observed when training a DLRM model (Naumov et al., 2019) on the **Criteo1TB** dataset (Lab, 2014) arises from the **large embedding tables** of the DLRM, which uses a vocabulary size of around 4 million entries (specifically $4194304$, Dahl et al. 2023). Frequent CPU-GPU transfers of model parameters–which include these large embedding tables–considerably slow down the algorithm, leading to more time spent transferring buffers than updating the model parameters.
> > >
> > > ---
> > > **References**
> > >
> > > Naumov, M., et al. Deep learning recommendation
> > > model for personalization and recommendation systems. 2019
> > >
> > > Lab, C. A. I. Criteo 1tb click logs dataset. 2014.
> > >
> > > Dahl, G. E., et al. Benchmarking neural network training algorithms. 2023.

---

### Official Review · Reviewer_A5Xg · 2025-03-13

**Overall Recommendation:** 4

**Summary:**

The authors benchmark weight averaging techniques using AlgoPerf, and find that without learning rate annealing it strongly accelerates training and generalization. It composes positively with learning rate annealing with small gains but cannot fully  replace annealing.

**Claims And Evidence:**

* Weight Averaging is generally helpful, especially in the absence learning rate annealing. Experimental evidence supports this.

* Weight Averaging Speeds Up Training. Experimental evidence supports this.

* Weight Averaaging + Learning Rate Annealing works best.  Experimental evidence supports this.

* Weight Avearaging cannot replace learning rate annealing.

**Essential References Not Discussed:**

It would be useful to discuss existing application of weight averaging a little bit, espeically in relation to robust finetuning [1] and OOD generalization [2]

[1]Wortsman, Mitchell, et al. "Robust fine-tuning of zero-shot models." Proceedings of the IEEE/CVF conference on computer vision and pattern recognition. 2022.
APA

[2]Arpit, Devansh, et al. "Ensemble of averages: Improving model selection and boosting performance in domain generalization." Advances in Neural Information Processing Systems 35 (2022): 8265-8277.

**Experimental Designs Or Analyses:**

Experiments using AlgoPerfm are well designed.

**Methods And Evaluation Criteria:**

The evaluation framework allows for extensive and diverse experiments, which is important for answering the research questions being asked in the paper.

**Other Comments Or Suggestions:**

It would be nice to clarify which dataset the figures are associated with in all cases. Eg. Figure 7.

I like the idea that learning rate schedules impede continual learning, since you decay to 0 at some point and restarting could be problematic from an optimization standpoint. Do the authors have any evidence that WA can be used for improved continual or transfer learning?

**Other Strengths And Weaknesses:**

Strengths: The paper is crystal clear in its findings, and this clarity is the basis for my recommendation of acceptance.

I like the finding that this weight averaging allows for access to a better model during training, useful for long duration training runs.

Weaknesses: The findings are not very surprising to the many using weight averaging. However, this is not a reason to reject IMO. We are lacking in clear presentation of simple ideas in this field.

**Questions For Authors:**

Why, in Figure 8, does annealing + weight averaging help whereas in figure 7b LAWA adds no benfit relative to the annealed model? Is it a different dataset, or difference in some other setting?

**Relation To Broader Scientific Literature:**

The scientific iteratture implictilty takes advantage of the findings of this paper extensively (wherever weight averaging is used). Formalizing them in a large scale study, however, presents the benfits clearly and is useful for distilling the implicit knowledge in the paper.

**Theoretical Claims:**

There are no Theoretical Claims.

---

> ### Author Rebuttal · Authors · 2025-03-31
>
> Thank you for your review and for your insightful comments. We appreciate your recognition of the value of our study! We respond below to specific comments.
>
> > It would be useful to discuss existing application of weight averaging a little bit, especially in relation to robust finetuning [1] and OOD generalization [2].
>
> Thank you for the additional references, we will include them in the revised manuscript.
>
> > It would be nice to clarify which dataset the figures are associated with in all cases. Eg. Figure 7.
>
> Thank you for spotting this! We will correct the labeling. Figure 7 is obtained by training a **Conformer** on Librispeech. We note that a similar effect can be observed on **other workloads** as well, and provide **additional results** [at this URL](https://anonymous.4open.science/r/ICMLE1C2/Short_LR.pdf) for comparing Weight Averaging against shorter LR schedules, [and at this URL](https://anonymous.4open.science/r/ICMLE1C2/Annealing_EMA.png) for testing the effect of EMA and LAWA when the learning rate is not annealed to zero. As suggested by Reviewers #egdu and #fFcZ, we will incorporate these additional experiments and considerations in the revised article to provide a broader view and strengthen our findings.
>
> > I like the idea that learning rate schedules impede continual learning, since you decay to 0 at some point and restarting could be problematic from an optimization standpoint. Do the authors have any evidence that WA can be used for improved continual or transfer learning?
>
> This is a very interesting question! Resuming training from a checkpoint obtained with standard cosine annealing can indeed present challenges, such as **potential forgetting** during the **re-warming phase** (Singh, V. et al., 2025). Therefore, it looks appealing to avoid annealing, and instead resume from the average model. We have not experimented with this point yet, but we think it would be an interesting direction for future work. An interesting question is how to optimally **resume training from the averaged model**, especially how to adapt the learning rate value in this scenario. Since averaging acts as a proxy of implicit learning rate decay, we hypothesize that using a smaller learning rate could be more beneficial for continued training from the average rather than restarting from the previous top learning rate. We believe this fits into the broader question of _how to better utilize the average for training_ (Kaddour et al., 2022; Defazio et al., 2024), a topic that remains underexplored but could offer significant benefits to the community.
>
> > Why, in Figure 8, does annealing + weight averaging help whereas in figure 7b LAWA adds no benfit relative to the annealed model? Is it a different dataset, or difference in some other setting?
>
> Figure 7b shows results on Librispeech Conformer, which, as reported in Figure 8, benefits only minimally when combined with LAWA. This small gains are not visible at the scale of the plot in Figure 7b.
>
> ### References
>
> Singh, V., et al. (2025). Beyond Cosine Decay: On the effectiveness of Infinite Learning Rate Schedule for Continual Pre-training. arXiv preprint arXiv:2503.02844.
>
> Kaddour, J. (2022). Stop Wasting My Time! Saving Days of ImageNet and BERT Training with Latest Weight Averaging. arXiv preprint arXiv:2209.14981.
>
> Defazio, A., et al. (2024). The Road Less Scheduled. arXiv preprint arXiv:2405.15682.

---

> > ### Comment · Reviewer_A5Xg · 2025-04-02
> >
> > Thank you for the rebuttal! I am satisfied and am raising my score accordingly.

---

### Official Review · Reviewer_5hDM · 2025-03-17

**Overall Recommendation:** 3

**Summary:**

This paper systematically explores two existing weight averaging techniques, EMA and LAWA, across the seven diverse tasks in the 2024 Algoperf competition. The authors find that weight averaging can either increase the final performance for a given compute budget or speed up training to specific performance threshold (by roughly ~20% on top of a well tuned baseline). They also explore whether weight averaging can replace learning rate decay but find that it can not. Rather it is always better to combine weight averaging with standard learning rate schedules (cosine in this paper) than using either method alone.

**Post-rebuttal update:** The authors have agreed to rephrasing edits which effectively address my actionable concerns. My conclusion remains that there is little to criticize about the paper aside the limited novelty of the methods and findings. I still think it is a nice reference and validation of the methods which is valuable. Overall these two factors roughly balance out to make this a borderline paper in my opinion.

**Claims And Evidence:**

Most of the claims are fine, but I have minor issues with some of them.

**We explore the relationship between averaging and learning rate annealing and show how to optimally combine the two to achieve better performance:** I do not agree that the paper shows how to **optimally** combine the two. What the paper does show is that combing standard weight averaging with standard learning rate schedules (as opposed to constant learning rates) is better than just using one of them. The paper does not show e.g. some new learning rate schedule that works better with EMA than standard techniques.

**(We) show that coupling WA with learning rate annealing yields optimal results— a novel finding to the best of our knowledge:** I somewhat disagree that this is a new finding. Practical works that use e.g. EMA on top of a learning rate schedule do so exactly because it works better than not using EMA (Karras 2312.02696 could be an example). Other works have shown that WA alone can not replace learning rate decay (e.g. Haegele 2405.18392). I think that the fact that the combination yields better results than one alone follows directly from these, although I am not sure if a single work has shown this.

**Essential References Not Discussed:**

None that I am aware of.

**Experimental Designs Or Analyses:**

Yes, I looked through the experimental setup and found no significant issues with it. Overall the experiments are convincing and well executed. Error bars are given for some results, further strengthening the results.

**Methods And Evaluation Criteria:**

Yes, evaluating weight averaging techniques on a competitive benchmark on top of very well tuned baselines is good. The scale and diversity of the benchmark is also convincing.

**Other Comments Or Suggestions:**

"Even for industrial-scale tasks like benchmark workloads, weight averaging reliably reduces compute costs by hundreds of GPU hours." - I think this sentence is a bit awkward. Only relative savings matter, 100s of GPU hours can range from very significant to very insignificant. The table below shows savings of roughly 100 hours, not 100s.

"even on second-order optimizers": There might be differing opinions on this but I think calling Shampoo a second order optimizer is a bit of a stretch (even if it is motivated by some very rough approximation of second order phenomena).

"coupling WA with learning rate annealing": I think "combining" would be clearer than "coupling" here. To me coupling signifies a tighter relationship than just using both methods. E.g. if you varied an EMA time horizon dynamically with the learning rate schedule or lowered the learning rate based on some signal from the EMA performance, that would better qualify as coupling.

On related work: I think some types of weight averaging were used earlier than you mention in deep learning. Maybe look into deep RL, the original transformer paper ("for the big models, we averaged the last 20 checkpoints"), and I think TensorFlow had support for some type of weight averaging very early on.

**Other Strengths And Weaknesses:**

The paper is well written and the figures are clear. The experiments are well done as previously mentioned.

There is little to criticize about the paper aside the limited novelty of the methods and findings. I still think it is a nice reference and validation of the methods which is valuable. Overall these two factors roughly balance out to make this a borderline paper in my opinion.

**Questions For Authors:**

No specific questions.

**Relation To Broader Scientific Literature:**

The paper does a good job of discussing related literature. I feel this paper primarily serves as a (decently diverse and large scale) validation of existing methods (standard weight averaging approaches) and observations about them (they improve performance, simple weight averaging can not replace lr schedules). I did not find the questions asked or the findings very novel or surprising. The empirical validation is still well done and can be of value to the community.

**Theoretical Claims:**

The paper doesn't make any theoretical claims.

---

> ### Author Rebuttal · Authors · 2025-03-31
>
> Thank you for your comments! We respond to specific comments below, but please let us know if you have any additional questions.
>
> > I do not agree that the paper shows how to optimally combine the two. What the paper does show is that combing standard weight averaging with standard learning rate schedules [...] is better than just using one of them.
>
> You are right—we do not explore *optimal* combinations of the two, but rather demonstrate that **combining** weight averaging with learning rate schedules outperforms using either alone. The broad scope of the benchmark makes a full optimality study expensive, but we believe this is still a useful result for practitioners. We will revise the manuscript to clarify this distinction. Thank you for pointing this out!
>
> > I somewhat disagree that this is a new finding. Practical works that use e.g. EMA on top of a learning rate schedule do so exactly because it works better than not using EMA.
>
> You are right, this is not a completely novel claim. Previous works indeed show that combining WA with learning rate schedules is beneficial. We will adjust the revised paper and clarify this point. We believe the impact of our work comes from demonstrating this at scale, unifying insights from previous work in a single competitive benchmark, and showing that WA alone does not replace learning rate decay across a wide range of settings.
>
> > Only relative savings matter, 100s of GPU hours can range from very significant to very insignificant.
>
> We agree that relative savings are important, and we will rephrase this for clarity. We did report the **15% saving** in GPU hours, which we believe to be significant for this benchmark, and will ensure this is emphasized more clearly in the revision.
>
> > I think calling Shampoo a second order optimizer is a bit of a stretch.
>
> We will clarify that while Shampoo is inspired by second-order principles, it may not strictly be a second-order optimizer.
>
> > "coupling WA with learning rate annealing": I think "combining" would be clearer than "coupling" here.
>
> Thank you for the useful suggestion. We agree that "combining" would be clearer than "coupling" and will make that adjustment in the revision.
>
> > On related work: I think some types of weight averaging were used earlier than you mention in deep learning.
>
> Thank you for the additional references. We are aware of this and will update the manuscript accordingly. We will add a paragraph in the Related Work section, citing earlier studies that predate SWA. Apart from the Transformer paper (Vaswani et al., 2017), we found that **other influential studies** such as Szegedy et al. (2016) and Merity et al. (2017) have also employed averaging schemes to improve model performance and reduce overfitting.

---

### Decision · Program_Chairs · 2025-05-01

**Decision:**

Accept (poster)

**Comment:**

This study explores the use of weight averaging (WA) across various neural network architectures and demonstrates its effectiveness in accelerating training and providing modest gains in generalization, using the AlgoPerf baseline. The results show that the advantages of WA are most prominent when the learning rate remains relatively high, and they diminish as the learning rate is annealed to zero. This suggests that WA is particularly beneficial during the more dynamic stages of training. The method appears to be generally applicable across architectures, though its impact may vary depending on the specific training regime.

Almost all reviewers agree that the paper merits acceptance, and after carefully reading the rebuttal and discussion, I tend to concur. Please incorporate the reviewers’ feedback into the final version of the paper, with particular attention to the following concerns:

1-  Incorporating additional experiments related to hyper-parameter tuning

2- Incorporate comparison with  other baseline optimizers ( as in your response to the reviewers )